

# Sexual dimorphism in the Arachnid orders

Callum J. McLean[1], Russell J. Garwood[2,3] and Charlotte A. Brassey[1]

[1] School of Science and the Environment, Manchester Metropolitan University, Manchester, UK
[2] School of Earth and Environmental Sciences, University of Manchester, Manchester, UK
[3] Earth Sciences Department, Natural History Museum, London, UK

## ABSTRACT

Sexual differences in size and shape are common across the animal kingdom. The study of sexual dimorphism (SD) can provide insight into the sexual- and natural-selection pressures experienced by males and females in different species. Arachnids are diverse, comprising over 100,000 species, and exhibit some of the more extreme forms of SD in the animal kingdom, with the males and females of some species differing dramatically in body shape and/or size. Despite this, research on arachnid SD has primarily focused on specific clades as opposed to observing traits across arachnid orders, the smallest of which have received comparatively little attention. This review provides an overview of the research to date on the trends and potential evolutionary drivers for SD and sexual size dimorphism (SSD) in individual arachnid orders, and across arachnids as a whole. The most common trends across Arachnida are female-biased SSD in total body size, male-biased SSD in relative leg length and SD in pedipalp length and shape. However, the evolution of sexually dimorphic traits within the group is difficult to elucidate due to uncertainty in arachnid phylogenetic relationships. Based on the dataset we have gathered here, we highlight gaps in our current understanding and suggest areas for future research.

## INTRODUCTION

Sexual dimorphism (SD), the difference in morphological, physiological and behavioural traits between males and females, is ubiquitous in nature. Common hypotheses to explain sex-specific divergence in body size and shape relate to sexual selection, intraspecific niche divergence and female fecundity pressures (*Shine, 1989*; *Andersson, 1994*). The first major step to understand the evolution of SD, however, is to document and describe the occurrence of sexually dimorphic traits in a wide range of species. Amongst vertebrates, for instance, the occurrence of SD is well documented. In mammals, it has been quantified in 1,370 species, representing around 30% of known mammalian species (*Lindenfors, Gittleman & Jones, 2007*). Datasets of similar size have been used to quantify SD in reptiles (1,341 species, *Cox, Butler & John-Alder, 2007*) and birds (*Owens & Hartley, 1998*). In contrast, the SD literature pertaining to invertebrates is more fragmented (*Abouheif & Fairbairn, 1997*), particularly within arachnids. Whilst a limited number of studies include large innterspecific datasets, their taxonomic breadth, relative to size of the group, pales in comparison to those in the vertebrate literature.

Corresponding author
Callum J. McLean,
callum.mclean@stu.mmu.ac.uk

Although such studies can highlight trends within specific groups, they provide only limited insight into trends across arachnids as a whole, primarily due to its diversity: the group comprises over 100,000 species (*Cracraft & Donoghue, 2004*).

Research into arachnid SD to date has largely focused on the spiders (Arachnida: Araneae). This is driven by interest in their conspicuous sexual size dimorphism (SSD), a subset of SD, which pertains solely to size differences in segments or body size between sexes. Interest in SSD in spiders stems from orb weaving spiders, which have the largest proportional weight difference between females and males of all studied land animals (*Foellmer & Moya-Larano, 2007*). Hence, research has probed the causes of this size disparity, and in particular the degree to which spiders follow Rensch's Rule, which states that if SSD is male-biased within a group, SSD will increase with the increased body size of a species; the converse is also true if SSD is female-biased in a group (*Rensch, 1950*). A focus on this question and group has left other arachnid orders relatively understudied, in terms of both SSD or SD in general.

The lack of study is unfortunate, as arachnids constitute an interesting group for learning more about SD, due to their wide range of morphologies, habitats and life histories. Indeed, SD is present in numerous forms throughout the arachnids, including the occurrence of exaggerated weapons (*Santos, Ferreira & Buzatto, 2013*), asymmetry (*Proctor, 2003*), extreme size dimorphism and other forms of polymorphism (e.g. Opiliones, Schizomida and Acari). The wide range of potential causes and expressions of dimorphism allow the influence of sexual selection and niche partitioning within the group to be assessed in great depth.

Recent advances make a review of SD in arachnids timely and important. Rigorous statistical testing has become commonplace in the last decade, with recent papers not only commenting on sexual differences, but also quantifying their significance (*Foellmer & Moya-Larano, 2007*; *Zatz et al., 2011*; *Santos, Ferreira & Buzatto, 2013*). Furthermore, high-resolution imaging has facilitated the study of smaller organisms, and the adoption of geometric morphometric techniques has allowed for sexual *shape* dimorphism to be quantified across a number of groups (e.g. humans, *Franklin et al., 2007*; reptiles, *Kaliontzopoulou, Carretero & Llorente, 2007*; spiders *Fernández-Montraveta & Marugán-Lobón, 2017*). Advances in phylogenetic methods have also made it possible to reconstruct the plesiomorphic state of sexually dimorphic traits, and the order of character acquisition in their evolution, thus providing novel data to help understand the drivers of SD (*Hormiga, Scharff & Coddington, 2000*; *Baker & Wilkinson, 2001*; *Emlen, Hunt & Simmons, 2005*).

In light of these new approaches, here we present the first review of SD across Arachnida. In particular, we have focused on collating data on the smaller arachnid orders, for which there is no pre-existing synthesis of SD. We begin by considering common methodological issues encountered throughout the arachnid SD literature. We move on to chart both SSD and shape dimorphism across eleven living orders, and touch on potential drivers in the evolution of sexually dimorphic arachnid traits. We conclude with a discussion of shared patterns in SD across Arachnida, and make suggestions for the direction of future research. As this review is of general interest to all researchers interested

in the development of SD and morphology, all arachnid-specific terms are defined or described as fully as possible.

## Considerations when studying sexual dimorphism in arachnids

Across the animal kingdom, metrics for quantifying SSD differ considerably between groups. In mammals, SSD is synonymous with dimorphism in body mass (*Weckerly, 1998*; *Lindenfors, Gittleman & Jones, 2007*). In contrast, in reptiles and fish SSD is often studied using body length (*Cox, Butler & John-Alder, 2007*; *Halvorsen et al., 2016*), in amphibians using snout-vent length (*Kupfer, 2007*) and in birds using wing or tarsus length (*Székely, Lislevand & Figuerola, 2007*). Mass is infrequently reported for arachnids. A primary challenge when reporting arachnid SSD is therefore identifying a linear reference character which reliably represents 'overall' body size in both sexes. Body length inclusive of opisthosoma, for example, may increase with feeding and is, to some degree, a measure of hunting success (as further outlined in sections 'Araneae' and 'Solifugae' below). As a result, total body size in arachnids is often taken as carapace length or width (*Weygoldt, 2000*; *Legrand & Morse, 2000*; *Pinto-Da-Rocha, Machado & Giribet, 2007*; *Zeh, 1987a*). However, carapace metrics can still be confounded by other shape variables (*Vasconcelos, Giupponi & Ferreira, 2014*; *Fernández-Montraveta & Marugán-Lobón, 2017*). For instance, the presence of unusual gland features in males of some spiders certainly modifies the shape of the carapace (*Heinemann & Uhl, 2000*). A number of potentially problematic reference characters are highlighted in the following review.

Sexual dimorphism in arachnids is often considered within the context of allometric scaling and support, or lack thereof, for Rensch's rule. Once a suitable reference character has been identified, advanced statistics can clarify when allometry is present, yet the choice of regression type bears consideration. Type-I (ordinary least squares) regression is recommended when variation in the dependent variable is more than three times that of the independent variable (*Legrende, 1998*), yet allometric studies of organismal morphology frequently do not meet this criterion. Applying Type-I models in instances where variance in the dependent and independent variables are similar can result in an underestimation of the regression coefficient (*Costa-Schmidt & Araújo, 2008*) and potentially hide allometric growth. Yet in situations when measurement error is low and measurement repeatability is very high, this underestimation is found to be negligible (*Kilmer & Rodríguez, 2016*). Furthermore, whilst many sexually dimorphic traits show positive allometry, sole focus on allometric scaling should be avoided. *Bonduriansky (2007)* found that many such characters (even those used as weapons in competition) scale isometrically, or with negative allometry, across a range of bird, fish and insect taxa. An emphasis on recording shape and overall size as opposed to just allometry is thus critical in determining the presence of SD.

When addressing the evolutionary drivers behind sexually dimorphic traits, it is important to avoid framing hypotheses around one sex (*Weygoldt, 2000*). For example, when studying SSD in orb-weaving spiders, the bulk of recent research has focused on the benefits of small body size in males (*Moya-Laraño, Halaj & Wise, 2002*;

*Foellmer & Moya-Larano, 2007*; *Grossi & Canals, 2015*). However, within a broader phylogenetic context, female gigantism is often considered more important in the development of size disparity (*Hormiga, Scharff & Coddington, 2000*). It is thus important to consider the advantages of differing morphologies from the perspective of both sexes.

Taxonomy may also be problematic, most notably when considering male polymorphism, as present in a number of arachnid groups (*Clark & Uetz, 1993*; *Gaud & Atyeo, 1996*; *Santos, Ferreira & Buzatto, 2013*; *Buzatto & Machado, 2014*). Assigning multiple male morphs to the corresponding female is challenging. Indeed, male polymorphism is likely to be more common than reported, but remains hidden due to the difficulties of placing differing morphs into the same species. This may further complicate the study of SD, particularly if sexes exhibit niche partitioning.

Finally, we note that caution is required due the inconsistent application of terminology within arachnology. Terms such as *setae* (referring to a stiff hair or bristle) and *flagellum* (a slender 'whip-like' appendage or body tagma) are used throughout arachnid literature to refer non-homologous structures. For example, the flagellum refers to a cheliceral appendage in solifuges and to a structure on the posterior opisthosoma in schizomids (*Harvey, 2003*). Conversely, homologous structures may be given different names across arachnids. The segments of the leg often carry different names between groups despite being homologous, and in the case of Amblypygi, homologous pedipalp segments are assigned differing names depending on author (*Weygoldt, 2000*). Where ambiguity in terminology exists, we provide descriptions of body segments where terminology alone may not describe position and form.

## Aim and survey methodology

A literature survey was conducted in Google Scholar using the scientific name of an arachnid order (e.g. 'Uropygi') and all common names ('whip scorpion', 'vinegaroon') and derivatives, with AND (the Boolean operator indicating that returned results should contain this and the subsequent term) then 'SD'. Google Scholar was chosen over other literature databases (e.g. Web of Science or Scopus) as the specified search terms may occur anywhere within the text, as opposed to only the title, abstract and keywords. Each returned paper was examined to determine if it contained pertinent information. Particular effort was made to identify and incorporate studies that quantified SD, especially those with statistical support. If no evidence of SD was provided, but a further citation was given, that citation was assessed. Additionally, arachnologists' personal paper collections were used to access further documents that did not appear in Google Scholar or citations. A full list of papers included, the form of dimorphism illustrated and the type of reporting used (qualitative vs. quantitative) is provided in the Supplementary Material. We highlight here that 'SD' refers to the condition in which males and females differ in their characteristics *beyond* primary sexual organs. The morphology of intromitent organs (penis in harvestmen and some mites, or pedipalps in spiders) and spermatophores, for example, is beyond the scope of this review.
## Standard figure abbreviations

Each section is accompanied with a figure charting general trends of SSD within the order. Figures follow a standard configuration: body parts coloured red indicate male-biased SSD, green indicates a female bias and purple mixed sex bias. Legs are numbered 1–4, chelicerae are marked 'C' and pedipalps are marked 'P'; male (♂) or female (♀) symbols denote SSD in overall body size. Other specific abbreviations are defined in figure captions. A plate of all SSD trend figures, for comparison across orders, is placed in the Supplementary Material.

## Acari

### Description and phylogeny

Acari, the subclass that contains mites and ticks, is the most speciose arachnid group with around 55,000 reported species (Zhang, 2011), although it is thought that this represents only a small fraction of a potential 1 million extant species (Walter & Proctor, 1999). Acari have colonised almost all terrestrial and marine environments and have also adopted modes of life including herbivory, predation, parasitism and scavengry (Vacante, 2015). Morphologically, Acari are distinct from the rest of the arachnids through their tagmosis, and the presence of a gnathosoma, a structure formed by the chelicerae, pedipalps and mouth, which form a functional unit separated from the rest of the body by a region of flexible cuticle. There are two major clades within Acari, the Parasitiformes and the Acariformes. They are differentiated morphologically by the stigmata arrangements; in Parasitiformes there are 1–4 dorsolateral or ventrolateral stigmata behind the coxa of leg II, which are absent in Acariformes (Vacante, 2015).

There is debate about monophyly of Acari, and multiple recent analyses have suggested that the two major clades are split making Acari polyphyletic. For example, Garwood et al.'s (2017) morphological phylogeny places Parasitiformes as the sister group to a clade including Acariformes and solifuges, and molecular phylogenies elsewhere agree with these results (Pepato, Da Rocha & Dunlop, 2010). However, other molecular studies place Acariformes as the sister group to pseudoscorpions, with this clade being the sister group to all other arachnids including Parasitiformes (Sharma et al., 2014). Earlier morphological phylogenies have also placed Acari as a sister group to Ricinulei (Lindquist, 1984; Shultz, 2007).

### Sexual dimorphism and potential drivers

The majority of literature concerning the SD in Acari focuses on the major acariform group Oribatida (Behan-Pelletier & Eamer, 2010; Behan-Pelletier, 2015a, 2015b). SD in feather mites has also been explored (Proctor, 2003). Within Orbatida, secondary sexual characters are generally considered rare (Behan-Pelletier & Eamer, 2010). SSD in overall body length is typically present but not pronounced in Orbatida: females are larger (Fig. 1), but male and female often overlap in size (Behan-Pelletier & Eamer, 2010). The most commonly SD is found in the dermal gland system (Behan-Pelletier & Eamer, 2010), with markedly different arrangements of the dermal porose areas reported between

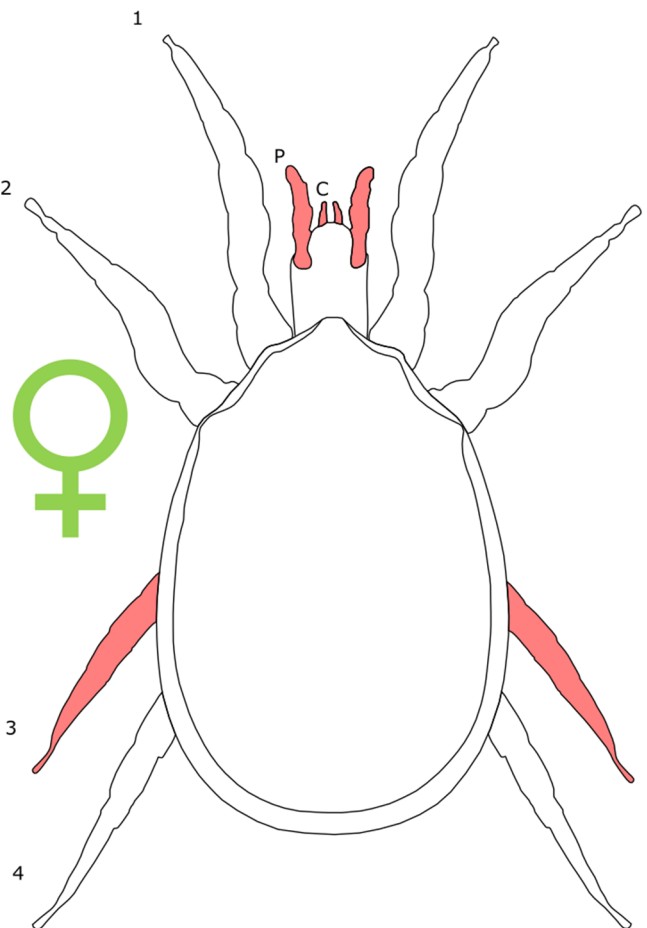

**Figure 1 Patterns of SSD across Acari.** See 'Standard Figure Abbreviations' for labelling guide.

sexes (*Norton & Alberti, 1997*; *Bernini & Avanzati, 1983*). These structures are used to spread sex hormones (*Norton & Alberti, 1997*) and male dermal glands can be associated with integumental structures on the carapace such as raised tubercles (*Behan-Pelletier & Eamer, 2010*).

Body shape dimorphism is reported in some mite species. In *Cryptoribatula euaensis*, the female carapace takes the semicircular form typical of the family Oripodidae, whereas the male carapace is pear shaped (*Behan-Pelletier & Eamer, 2010*). The arrangements of plates comprising the exoskeleton can also differ between sexes in Oribatida, as can the occurrence of setae and other integumental structures (*Behan-Pelletier & Eamer, 2010*; *Behan-Pelletier, 2015b*). In extreme cases, the idostoma, the body segment that attaches to the legs, can even be bifurcated (*Proctor, 2003*). In several groups of feather mites, body shape is non-symmetrical across the sagittal plane in males (*Proctor, 2003*; *Proctor & Knee, 2018*). In those taxa characterised by male polymorphism (where males occur in multiple morphotypes, often reflecting different mating strategies; e.g. *Radwan, 1993*; *Ra'Anan & Sagi, 1985*; *Tsubaki, 2003*), males can be both symmetrical and asymmetrical (*Proctor, 2003*).

The evidence for SSD in leg length is limited, and appears to favour males. In two species of *Ameronothrus*, leg length exceeds body width in males, whilst the opposite is true for females (*Søvik, 2004*; *Behan-Pelletier & Eamer, 2010*). This may not represent true SSD in leg length as females also have a larger body size in this species (*Søvik, 2004*). Male-bias SSD in the third leg length has also been documented (*Gaud & Atyeo, 1996*). Furthermore, male legs are often modified with flanges, lobes, leg clamps, adanal discs or pincers (*Proctor, 2003*). Setal arrangement also varies between sexes, with male orbatids having modified setae on the legs that are absent in females (*Behan-Pelletier & Eamer, 2010*; *Behan-Pelletier, 2015b*). Within the gnathosoma, male pedipalps are enlarged relative to female conspecifics. In some species of Astigmata, males also have pedipalp branches unseen in females of the same species, and in the most extreme cases the pedipalps appear antler-like (*Proctor, 2003*). Chelicerae are also enlarged in some male feather mite species (*Proctor, 2003*). There are a number of prodorsal modifications present exclusively in males of some acarid species, which are hypothesised to help the male push female towards their spermatophore (Behan-Pelleiter & Eamer, 2015b). This suggest the influence of sexual selection acting through a form of sexual coercion.

Potential drivers for dimorphism in Acari are difficult to determine given the relative lack of information on life history. A correlation between habitat and SD has been discussed in Oribatida, as the majority of sexually dimorphic species occur in non-soil environments (*Behan-Pelletier & Eamer, 2010*), despite Acari as a whole being more speciose in the soil (*Behan-Pelletier & Eamer, 2010*). Likewise, SD in the glandular system has been linked to habitat, as sex pheromones emitted from dermal glands are potentially more important for attracting a mate in drier environments (*Norton & Alberti, 1997*). Dimorphism in the nymphs of Kiwi bird (Aves: Apterygiformes) mites has also been attributed to their environment, with males living in feathers and females living in cutaneous pores, being one of the few unequivocal examples of niche partitioning between species in arachnids (*Gaud & Atyeo, 1996*).

Mating has been hypothesised to play a role in the elaboration of the third legs of male feather mites. The lobes, flanges and setae on the legs potentially help males to align with the female spermaduct opening (*Gaud & Atyeo, 1979*), and sexual selection could drive the development of these modifications. Elsewhere, heteromorphic 'fighter' males of *Caloglyphus berlesei* use their enlarged third legs to kill rival males (*Radwan, 1993*) and monopolise females. In contrast, non-fighter males, which do not kill off rival males, are more successful in larger colonies under laboratory conditions (*Radwan, 1993*); factors such as population density may therefore influence mating behaviour and thus sexual- and male-dimorphic morphology.

Research into SD among mites and ticks has thus far been limited in taxonomic scope. Advances in high-resolution 3D imaging could assist future research into SD in smaller mites. We believe mites present an interesting study organism for interrogating the interplay between morphology and mating strategies. For example, many oribatid mites can and do reproduce via parthenogenesis (*Behan-Pelletier & Eamer, 2010*); the extent to which species that reproduce in this manner exhibit SD is as yet unknown.

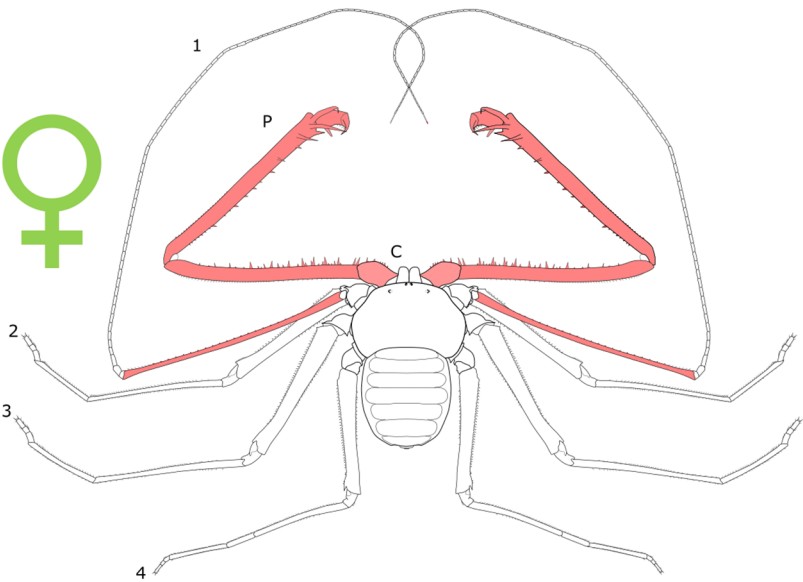

**Figure 2 Patterns of SSD across Amblypygi.** Though carapace has been found to be statistically wider in males in *Charinus jibaossu* relative to carapace length, suggesting a larger carapace overall, it is not highlighted here due its wide consideration as a reference character for overall body size, which is thought to favour females. See 'Standard Figure Abbreviations' for labelling guide.

## Amblypygi

### Description and phylogeny

Amblypygi, or whip spiders, are an arachnid order comprising ca. 220 species (*McArthur et al., 2018*). Amblypygids live in tropical regions, preferring rainforests and caves and are obligate predators (*Weygoldt, 2003*). Members of the order have a distinct morphology, their most recognisable trait being raptorial pedipalps exceeding twice the individual's body length in some taxa (*Weygoldt, 2000*). Amblypygids also possess antenniform first legs known colloquially as whips, which bear sensory devices thought to allow mechano- and chemoreception (*Igelmund, 1987*). Amblypygi also lack a terminal flagellum, which differentiates them from the other two orders that comprise the clade Thelyphonida, Uropygi and Schizomida (following the International Society of Arachnology). Recent morphological and molecular phylogenies consistently place amblypygids in a clade with thelyphonids (*Shultz, 2007*; *Garwood & Dunlop, 2014*; *Sharma et al., 2014*; *Garwood et al., 2017*).

### Sexual dimorphism and potential drivers

Female-biased SSD in overall body size, as measured by carapace width, is common across Amblypygi (*McArthur et al., 2018*), potentially relating an increased capacity for egg production at larger body sizes (*Armas, 2005*) via fecundity selection. Male-biased SSD in pedipalps is widespread across the group, but the level of dimorphism varies greatly between species (*McArthur et al., 2018*; Fig. 2). In *Damon variegatus* and *D. gracilis*, pedipalpal tibia length scales similarly in males and females across early instars. However, after the fourth nymphal stage, the pedipalpal tibia displays greater positive allometry

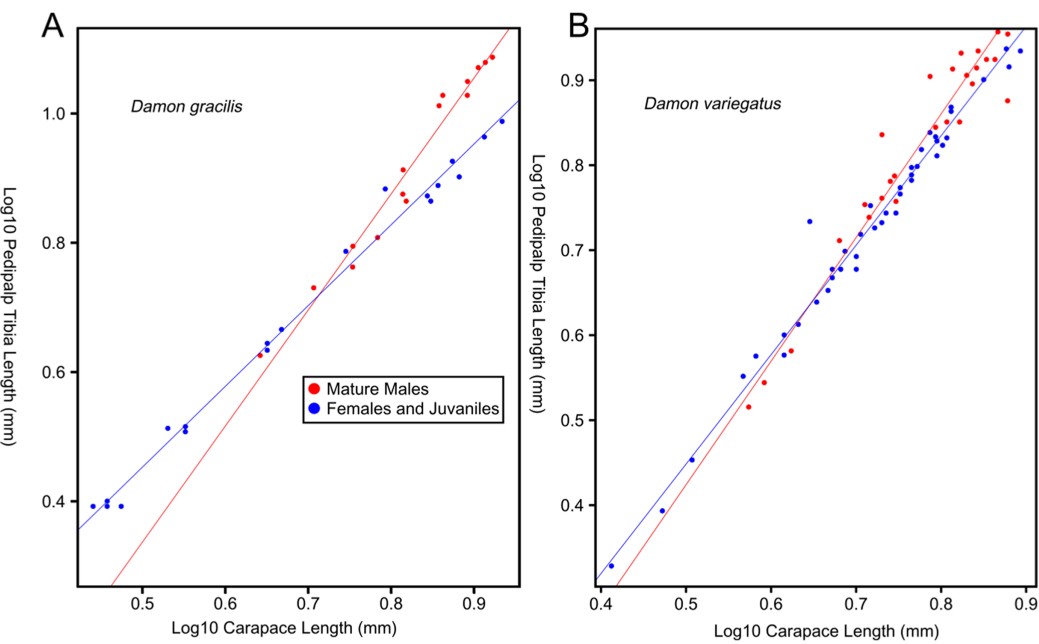

**Figure 3 Relationship between log pedipalp tibia length and log carapace length.** Relationship between log pedipalp tibia length and log carapace length (modified from *Weygoldt, 2000*). Regression analysis was re-run with a type two regression; against the H0 that the two rates of allometric growth are equal $p$ = <0.001 for *Damon gracilis* (A), $p$ = 0.031 for *Damon variegatus* (B).

relative to carapace length in males (*Weygoldt, 2000*; Fig. 3). A similar growth pattern has been identified in the pedipalpal tibia of *Phrynichus deflersi arabicus* (*Weygoldt, 2003*), *Phrynus marginemaculatus* and *Heterophrynus batesii* (*McArthur et al., 2018*). Male-bias SSD in pedipalpal length has also been observed in adults of several other species (e.g. *Charinus mysticus* and *Sarax huberi*), albeit with smaller sample sizes (*Vasconcelos, Giupponi & Ferreira, 2014*; *Seiter, Wolff & Hoerweg, 2015*). Pedipalpal spines may also be sexually dimorphic in Amblypygi. Both male and female adult *Euphrynichus bacillifer* possess spines transformed into rounded apophyses, yet these are both larger and carry more glandular pores in males. *Phrynichus exophthalmus* also has a blunt apophasis on the pedipalp in males but not in females (*Weygoldt, 2000*). The function of the apophyses and their associated glandular pores remains unclear (*Weygoldt, 2000*). SD in the number of pedipalpal spines has also been reported in *Charinus jibaossu* (*Vasconcelos, Giupponi & Ferreira, 2014*).

Recent work has suggested that territorial contest could be a driving force behind pedipalp SSD in amblypygids. Field observations of *Phrynus longipes* have found that the majority of territorial contests (82.8% in trials) are decided purely via display (*Chapin & Reed-Guy, 2017*). In these trials, the winner was always the individual with the longest pedipalpal femur length, creating a selective pressure for longer pedipalps. However, investment in pedipalps is a high-risk strategy, as in those interactions that escalate to contest and cannibalism, the winner is best predicted by body size (*Chapin & Reed-Guy, 2017*). A recent study has also reported that the level of SSD across

amblypygid species decreases with distance from the equator (*McArthur et al., 2018*). This may indicate climatic controls on mating strategy, as has been demonstrated in Opiliones (*Machado et al., 2016*), but further research is required.

The antenniform first pair of legs has also been observed to be dimorphic in a number of species across the group, and statistically demonstrated in *P. marginemaculatus* and *H. batesii* (*McArthur et al., 2018*). Male–male confrontation follows a common pattern across Amblypygi: initially, males 'fence' by turning side-on to one another and repeatedly touching antenniform legs, before unfolding their pedipalps, turning face on and charging (*Weygoldt, 2000*). Males also use whips to display to females and touch the female's body before mating (*Weygoldt, 2000*). Whip legs are also thought to have chemoreceptive functions (*Weygoldt, 2000*) that could hypothetically aid in mate search, although no link has yet been draw between whips and the ability to locate potential mates. It would therefore appear that SSD in whip length is driven by sexual selection though male contest and potentially female mate choice via pre-copulatory courtship.

Body segments can also show dimorphism, although it is rare in the group (*Weygoldt, 2000*). Shape dimorphism can be observed in *C. jibaossu*, with the male having wider carapace relative to length than females (*Vasconcelos, Giupponi & Ferreira, 2014*). *McArthur et al. (2018)* also reported widespread female biased dimorphism in carapace width, although it was being considered a proxy for overall body size. In *Damon medius* and *D. variegatus*, females possess a pleural fold along the ventrolateral and posterior opisthosomal margins; in ovigerous females, this fold surrounds the eggs to form a brood pouch (*Weygoldt, 2000*). On the underside of the opisthosoma, females of some species in the family Phrynichidae possess an area of red-gold hair around the posterior margin of the genital opening, that is, otherwise absent in males (*Weygoldt, 2000*).

Sexual dimorphism in amblypygids is understudied relative to the larger arachnid orders. Several publications report little or no dimorphism within species (*Rahmadi, Harvey & Kojima, 2010*; *Giupponi & Kury, 2013*). By necessity, these rely on small sample sizes: amblypygids are seldom seen in large numbers in the wild and are thus difficult to collect (*Weygoldt, 2000*). As a result, quantitative tests are either not possible, or low in statistical power. Furthermore, subtle sexual character dimorphism (e.g. differences in pedipalpal dentition) are easily overlooked in studies that rely on linear metrics. Future work will benefit from revisiting existing amblypygid collections, and utilising advances in imaging and 3D morphometrics.

## Araneae

### Description and phylogeny

Araneae—or spiders—are the archetypal arachnid, and the order comprises over 47,500 species (*World Spider Catalog, 2018*). Spiders are found in almost all terrestrial habitats. They are always predatory and possess weapons that are absent in other arachnids, such as the ability to administer venom via the chelicerae, and the ability to spin silk using opisthosomal spinnerets. Araneae are members of a clade containing

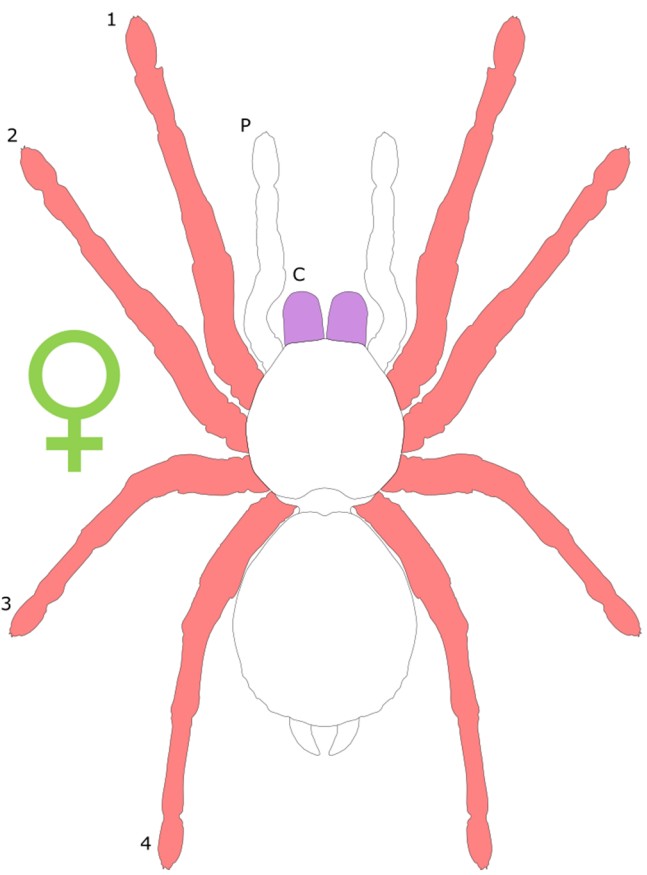

**Figure 4** **Patterns of SSD across Araneae.** See 'Standard Figure Abbreviations' for labelling guide.

Amblypygi and Uropygi; their sister group is thought to be either Amblypygi (*Wheeler & Hayashi, 1998*) or Pedipalpi as a whole (a clade comprising Amblypygi, Uropygi and Schizomida; *Shultz, 2007*; *Sharma et al., 2014*; *Garwood et al., 2017*).

### Sexual dimorphism and potential drivers

Spiders are typically characterised by female-biased SSD, with females outweighing male conspecifics by up to two orders of magnitude (*Foellmer & Moya-Larano, 2007*; Fig. 4). In web-building spiders, female body length frequently exceeds that of males (*Head, 1995*; *Vollrath, 1998*) and can be twice that of males (*Hormiga, Scharff & Coddington, 2000*). Extreme female-biased SSD is particularly prevalent in the families Thomisidae and Araneidae (*Hormiga, Scharff & Coddington, 2000*). The bulk of research concerning SD in spiders has concentrated on the prevalence of female-bias SSD and the potential driving factors underlying such extremes in total body size. The so-called 'giant females vs. dwarf males' controversy (*Coddington, Hormiga & Scharff, 1997*) has been discussed in detail elsewhere (see *Moya-Laraño, Halaj & Wise, 2002*; *Foellmer & Moya-Larano, 2007*), and is not covered further in the present review. Likewise, the degree to which total body size SSD in Araneae is consistent with the predictions of Rensch's rule has been the subject of considerable study. The current consensus appears to be that SSD actually

increases with body size in spiders characterised by female-bias SSD (*Abouheif & Fairbairn, 1997*; *Prenter, Elwood & Montgomery, 1999*) counter to Rensch's rule, with male and female body size showing relatively uncorrelated evolution (*Foellmer & Moya-Larano, 2007*). Furthermore, interesting exceptions to female-biased SSD do exist; for example, the aquatic spider *Argyroneta aquatica* displays male-bias SSD in total body length (*Schütz & Taborsky, 2003*). *Linyphia triangularis* also subverts the general trend with males having wider cephalothoraxes than females (*Lang, 2001*), and male of the wolf spider *Allocosa brasiliensis* are larger than females in cephalothorax length (*Aisenberg, Viera & Costa, 2007*).

It should be noted that the above studies consider body size SSD within the context of body length (*Head, 1995*; *Elgar, 1991*). Body length is subject to change based on hunting success, resulting in potential overestimation of female body size in particular, as they tend to feed more over their life span (*Legrand & Morse, 2000*). Carapace width is unaffected, however, and remains roughly constant within an instar stage (*Legrand & Morse, 2000*), and may therefore become the preferred metric in future studies of SSD in spiders. However, the use of carapace width as a predictor of body size can also be problematic in instances when the prosoma itself shows SD. In *Donacosa merlini* (Lycosidae), geometric morphometric analysis found the male carapace to be statistically wider and more anteriorly protruding than that of the female relative to overall size (*Fernández-Montraveta & Marugán-Lobón, 2017*). The authors also report differences in the relative sizes of the prosoma and opisthosoma, which is suggested to result from the larger female opisthosoma creating a fecundity advantage by stowing more eggs, with other studies finding strong correlation between female carapace size and clutch size (*Pekár, Martišovà & Bilde, 2011*; *Legrand & Morse, 2000*). Statistically significant SSD in carapace width and height is also present in the linyphiid *Oedothorax gibbosus* (*Heinemann & Uhl, 2000*). This results from a large gland located within the male cephalothorax that supplies a nuptial secretion to females during courtship (*Vanacker et al., 2003*). The presence of this gland is also male dimorphic, and males of the morph that lacks the gland have a smaller carapace. This likely indicates a divergence in male mating behaviour (*Heinemann & Uhl, 2000*).

Sexual dimorphism in the pedipalps of spiders must be considered with caution. Within Araneae, the male pedipalp is principally adapted to transfer spermatophores to the female reproductive tract. As such, they effectively function as genitalia, and sex-based differences are examples of 'primary' SD. Unlike other arachnid groups, secondary SD in the pedipalps is rare in spiders. However, males of some burrowing wolf spiders, namely *Allocosa alticeps* and *A. brasiliensis*, possess palpal spines that are absent in conspecific females (*Aisenberg et al., 2010*). Contrary to other burrowing wolf spider taxa, males of these two species burrow while females engage in active mate search, and modifications to male pedipalps are thought to improve burrowing performance (*Aisenberg et al., 2010*).

Male-bias SSD in leg length relative to total body size is commonly observed in Araneae (*Foellmer & Moya-Larano, 2007*). Hypotheses for its adaptive significance fall into two broad categories: locomotion and display. Increased leg length has been linked to a

theoretical increase in climbing and bridging speed (*Grossi & Canals, 2015*), whilst other authors have argued for the role of sexual cannibalism in imposing a selective pressure towards longer legs to aid in escape (*Elgar, Ghaffar & Read, 1990*). Male-bias SSD in leg length has also been correlated with active mate searching, because male wolf spiders involved in active mate searching possess longer legs relative to those of females (*Framenau, 2005*). Interestingly, in wolf spider taxa in which females actively search for mates, female-biased SSD in leg length becomes common, though examples of this reverse in SSD bias are thought to be uncommon (*Aisenberg et al., 2010*).

In contrast, the legs of male salticids (jumping spiders) are commonly elongated and ornamented with setae for the purpose of display. Male peacock spiders possess elongated third legs relative to females, which are used in a ritualised courtship dance, often tipped with white bristles (*Girard & Endler, 2014*). Males of *Diolenius phrynoides* also show extreme lengthening of the first legs, which are adorned with ridges of setae on the tibia unlike those of the female; again for use in display (*Peckham & Peckham, 1889*). Elongation of the forelegs in male wolf spiders has likewise been related to courtship (*Kronestedt, 1990*), supported by the presence of heavily pigmented bristles in the male *Schizocosa ocreata* (*Scheffer, Uetz & Stratton, 1996*). This species displays 'drumming' behaviour, where males beat their legs against the ground in order to attract prospective mates. In situations where the substrate hinders the transmission of the drumming, females prefer males with intact bristles, providing evidence they also play a visual role in courtship displays (*Scheffer, Uetz & Stratton, 1996*). Intersexual contest could also drive dimorphism in the legs of some species. Fighting behaviour using the legs as weaponry has been observed between males in the genera *Modisimus* and *Blechroscelis*, with males typically using their legs to push against the opponent (*Eberhard-Crabtree & Briceño-Lobo, 1985*).

Spider chelicerae are also characterised by SSD, although the direction of dimorphism is less consistent than in the pedipalps or legs. Unlike isometric females, male *Zygoballus rufipes* chelicerae exhibit positive allometric growth in length relative to carapace length, with the resultant enlarged chelicerae in adult males thought to be involved in courtship display (*Faber, 1983*). Taxa in which males present nuptial gifts to prospective mates are also characterised by male-bias SSD in absolute cheliceral size, although the structures do scale with isometry (*Costa-Schmidt & Araújo, 2008*). In wolf spiders though, female chelicerae have been reported to be statistically larger than males (*Walker & Rypstra, 2002*). Increased dentition on the chelicera base is also seen in males of some species (*Peckham & Peckham, 1889*), but the purpose of this is unclear. Given that chelicerae are used in male–male competition and that fighting success is a good predictor of mating success in spiders (*Rovner, 1968*; *Watson, 1990*), intrasexual selection may also underlie the hyper-allometric growth of male chelicerae (*Funke & Huber, 2005*).

Alternatively, SSD in *Myrmarachne palataleoides* chelicerae has been attributed to differing forms of prey capture between males and females, in which the relatively longer chelicerae of males are used to spear and dispatch prey in the absence of venom, which appears only in female conspecifics (*Pollard, 1994*). Dimorphism in some wolf spider

chelicerae has also been correlated to dietary differences between the sexes, in turn relating to their respective reproductive roles. Females are known to catch significantly more prey items, and show statistically significant female-biased dimorphism in cheliceral paturon (the segment housing chelicerae muscles, adjacent to the fang) length, width and fang width (*Walker & Rypstra, 2002*). Little evidence of habitat niche divergence between sexes exists, indicating female-biased SSD in chelicerae was likely a response to increased feeding induced by the energetic cost of rearing young (*Walker & Rypstra, 2002*). Female-biased SSD in chelicerae in the ant-eating spider *Zodarion jozefienae* also appears to be related to trophic niche partitioning. Due to the increased energetic demands of fecundity, females prey on larger morphs of *Messor barbarous* ants than males (*Pekár, Martišovà & Bilde, 2011*).

Sexual body character dimorphism in ornamentation, patterning and colouration are also common across Araneae. Female orb-weaving spiders have a highly ornamented carapace comprising spines and bright colours, which are otherwise lacking in males (*Peckham & Peckham, 1889*). In the spiny orb-weaving genera *Micrathena* and *Chaetacis*, elongate abdominal spines have evolved independently in females on eight separate occasions, and may exist as anti-predator structures for the usually larger and thus more conspicuous females (*Magalhaes & Santos, 2012*). In salticids, however, males are characterised by increased colouration. Male *Habronattus decorus*, for example, possess a purple opisthosoma and brighter colours on the legs and prosoma than their black and white female counterparts do (*Peckham & Peckham, 1889*). Further SD is visible when some taxa are viewed under ultraviolet (UV) light. For example, only male *Cosmophasis umbratica* have body parts that reflect UV light (*Lim & Li, 2006*). Salticids are capable of detecting light well within the UV spectrum (*Peaslee & Wilson, 1989*), and female *C. umbratica* exhibit a preference for UV-reflecting mates as opposed to those with UV-reflecting capabilities masked (*Bulbert et al., 2015*). Such research highlights the importance of considering other potential modalities for dimorphism that are less obvious to the human observer (*Huber, 2005*).

In Theraphosidae, commonly known as tarantulas, SD occurs in both the size and composition of urticating setae, which are hairs expelled when the spider is threatened, causing respiratory distress in vertebrates (*Bertani & Guadanucci, 2013*). Longer urticating setae have been reported in males compared to females of numerous species, and statistically significant differences identified in *Avicularia avicularia* (*Bertani & Guadanucci, 2013*). Setae composition is also sexually dimorphic, with females of three different genera possessing only Type-I setae, which are shorter hairs thought to defend against other invertebrates (*Bertani & Guadanucci, 2013*). In contrast, males possess both Type-I and Type-III setae, the latter being a longer seta used to ward off vertebrates. Differences in setal composition may relate to the males' requirement to search for mates, placing them at greater risk of encountering vertebrate predators (*Bertani & Guadanucci, 2013*).

Spiders are by far the most-studied arachnid order in terms of SD, and particularly SSD. Research in this group has benefitted from a number of novel approaches, including advanced imaging techniques (e.g. studies in UV reflectivity and histological sectioning),

kinematics and biomechanical testing. The application of such techniques to other arachnid orders may prove useful in future research. Additionally, sample sizes are often far in excess of those generated on non-Araneae arachnids.

## Palpigradi

### Description and phylogeny

Palpigradi, or micro-whip scorpions, are one of the least studied arachnid orders (see Supplementary Table). There are 78 extant species that are primarily found in leaf litter and caves across the tropics (Condé, 1996; Harvey, 2003). Diagnostic features include a long, segmented terminal flagellum coupled with tri-segmented chelicerae (Harvey, 2003). Moreover, all species are very small, and typically average 1–1.5 mm in total length (Ax, 2000). The order Palpigradi has been placed in Tetrapulmonata with Amblypygi, Araneae, Uropygi and Schizomida (Shultz, 1990; Wheeler & Hayashi, 1998), but also as a sister group to different groups, including Acariformes (Van Der Hammen, 1989; Regier et al., 2010), solifuges (Giribet et al., 2002) or the rest of Arachnida (Shultz, 2007). The most recent studies have placed Palpigradi as the sister group to Parasitiformes (Sharma et al., 2014) or to the remaining arachnids (Garwood & Dunlop, 2014; Garwood et al., 2017).

### Sexual dimorphism and potential drivers

To date, SSD in overall body size has not been reported in Palpigradi (Fig. 5), and expression of SD occurs predominantly in setal arrangements. In *Eukoenenia chilanga*, males have more setae on the opisthosomal sternites, ventral sclerotized plates making up opistosomal segments X and XI (Montaño-Moreno & Francke, 2013). The number of setae also differs on other opistosomal segments, with male *E. mirabilis* possessing 31 setae on sternite VI compared to six or seven in the female (Condé, 1991). Setae are generally thicker and more cylindrical in males (Barranco & Mayoral, 2007; Souza & Ferreira, 2012).

Dimorphism in the palpigrade glandular systems have also been observed. In *E. lawrencei*, females possess three large glandular masses that protrude under segment VII compared to two glands in the males (Condé, 1991). The extra glands in females may play a role in reproduction (Condé, 1991), though this is not elaborated on. The degree to which the above differences are statistically significant remains untested, however, and previous studies are limited by small sample sizes.

Further work is needed for the patterns and drivers of SD in Palpigradi to be understood. As far as we are aware, the mating habits of Palpigradi have never been reported, and relatively little is known of their ecology and behaviour. An improved understating of the mating and courtship behaviors will prove important for identifying the potential drivers of observed dimorphism.

## Pseudoscorpiones

### Description and phylogeny

Pseudoscorpions, occasionally referred to as book scorpions (or sometimes false scorpions), are represented by over 3,300 species (Garcia et al., 2016). Members of the order are found in a wide range of terrestrial environments, typically in the tropics and
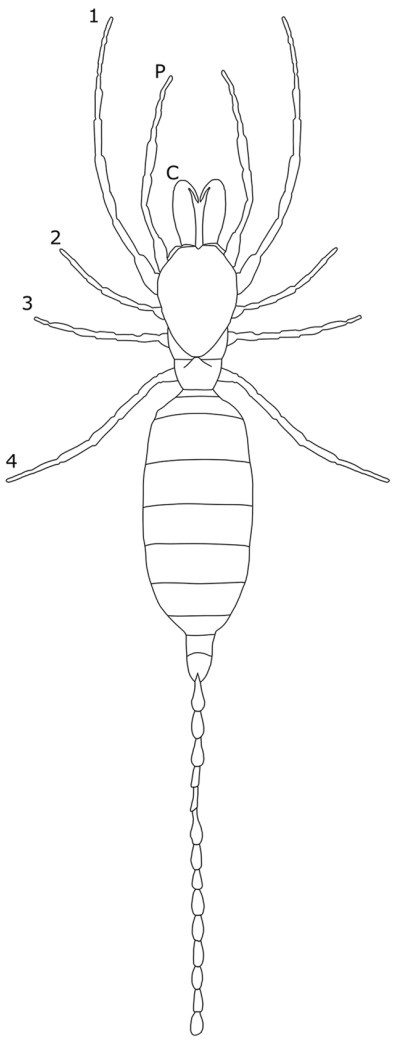

**Figure 5 Patterns of SSD across Palpigradi.** See 'Standard Figure Abbreviations' for labelling guide.

subtropics, although occasionally as far north as arctic Canada (*Muchmore, 1990*). Pseudoscorpions appear superficially similar to scorpions, possessing pedipalpal claws and a segmented opisthosoma, although they lack the tail and telson seen in true scorpions. They also differ from scorpions in size; the largest pseudoscorpion reaches only 12 mm in total body length (*Beier, 1961*) yet most measure approximately one mm (*Schembri & Baldacchino, 2011*). Some morphological studies place pseudoscorpions as the sister group to scorpions (*Pepato, Da Rocha & Dunlop, 2010*; *Garwood & Dunlop, 2014*; *Garwood et al., 2017*) and others to solifuges (*Legg, Sutton & Edgecombe, 2013*; *Giribet et al., 2002*; *Shultz, 2007*). Molecular studies, in contrast, have placed them as the sister group to acriform mites (*Sharma et al., 2014*).

### Sexual dimorphism and potential drivers

Overall body size dimorphism is well documented in pseudoscorpions. In Cheiridioidea, a a large superfamily containing the well-studied Chernetidae (*Murienne, Harvey &*

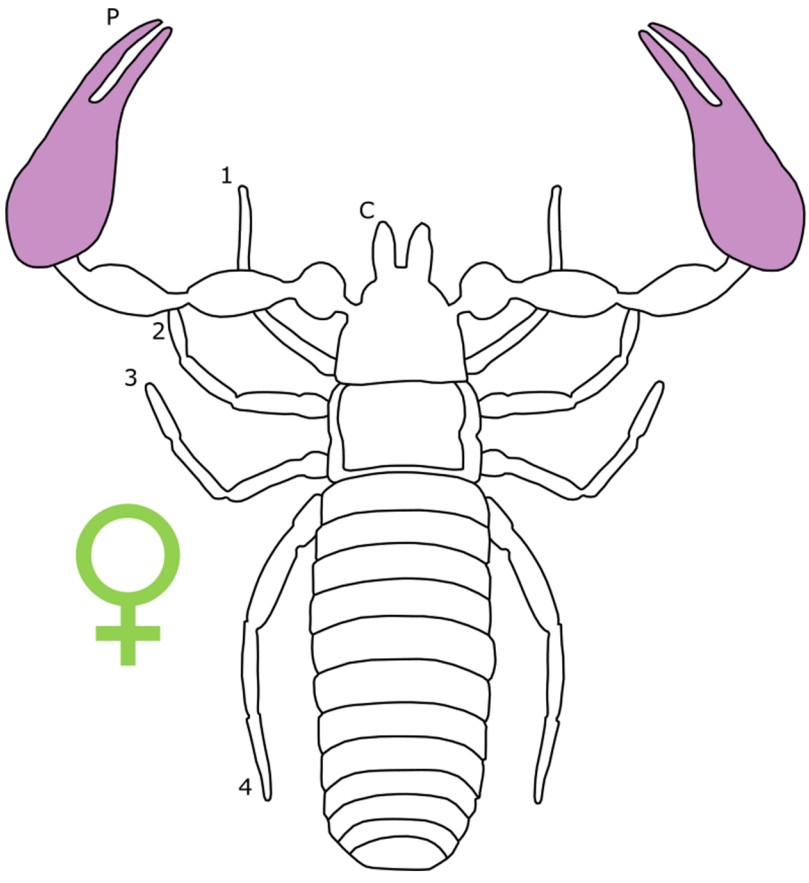

**Figure 6 Patterns of SSD across Pseudoscorpiones.** See 'Standard Figure Abbreviations' for labelling guide.

*Giribet, 2008*), males are consistently smaller than females, measured by carapace length (*Zeh, 1987a*). In fact, *Zeh (1987a)* notes that male-biased SSD is rare in Chernetidae, finding just eight species that exhibit reverse SSD in the 45 that were studied (*Zeh, 1987a*).

Sexual size dimorphism in pseudoscorpion pedipalps is present in a number of species. Males in the family Chernetidae typically have larger pedipalpal claws than females (*Zeh, 1987a*, *1987b*; Fig. 6). This is highly variable however: male claw silhouette area ranges from 60 to 150% of that in females (*Zeh, 1986*; Fig. 7). Furthermore, the direction and extent of dimorphism can vary significantly within a genus. It is not uncommon to find both strong male-biased and female-biased SSD in claw size within a genus (*Zeh, 1987b*; Fig. 7). Regression analysis also reveals that the SSD in male claws seems to increase relative to female body size (*Zeh, 1986*). However, we note that this trend is not normalised to body size. Thus, whilst absolute difference in claw size increases, this could be primarily due to changes in body size.

Several pseudoscorpion groups engage in 'pairing', a ritualised dance in which the male grasps the female's pedipalpal claws before depositing a spermatophore (*Weygoldt, 1966*). *Zeh (1987a)* has suggested pairing may be a major control on dimorphism, particularly in pedipalpal claws. Furthermore, male–male aggression has been correlated to

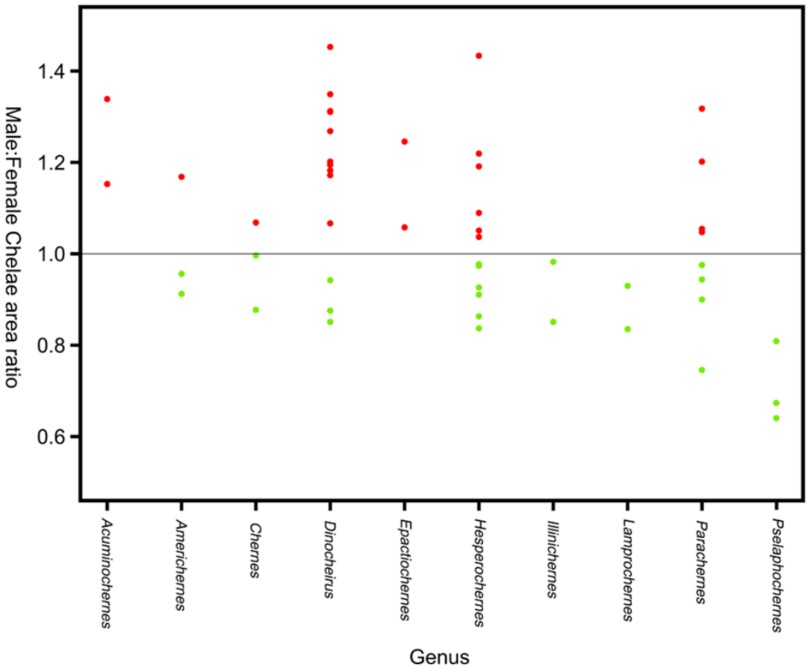

**Figure 7 Patterns of sex bias in pedipalp claw SSD in Psuedoscorpions.** Patterns of sex bias in pedipalp claw SSD in Psuedoscorpions, red dots indicate male bias, green is female-biased. Modified from *Zeh (1987a)*.

SSD in pedipalps. Male pseudoscorpions often fight each other using the pedipalpal claws (*Weygoldt, 1966*; *Thomas & Zeh, 1984*), and experimental work suggests chela size, not body length, is a good predictor of the victor in such contests. Notably, it has also been reported that males with larger chelae produce more spermatophores than those with smaller chelae, suggesting they may have greater mating success (*Zeh, 1986*). A weak but significant relationship between the level of SSD and population density in Chernetidae has been reported. SSD was also found to be more pronounced in specimens taken from nesting areas (*Zeh, 1986*).

Sexual dimorphism in pseudoscorpions is therefore well documented. Studies have included extensive statistical testing on morphometric characteristics, and the selective pressures driving SD are comparatively well understood. SSD has been particularly well described in Chernetidae, yet substantially less is known of other pseudoscorpion families. This is where significant gaps in the current body of knowledge lie.

## Opiliones

### Description and phylogeny

Opiliones, commonly known as harvestmen or daddy long-legs, are the third largest arachnid order comprising over 6,500 species (*Kury, 2013*). The greatest diversity of harvestmen is in the tropics, though their range stretches into the high-latitudes (*Pinto-Da-Rocha, Machado & Giribet, 2007*). A common characteristic of harvestmen is the second pair of legs, which carry both mechano- and chemoreceptors (*Willemart & Chelini, 2007*). Synapomorphies of the group include the position of the gonopore,

the presence of a penis or spermatopositor for direct copulation, and the presence of repugnatorial glands (*Pinto-Da-Rocha, Machado & Giribet, 2007*). The majority of recent phylogenetic analyses have placed Opiliones as the sister group to a clade comprising pseudoscorpions and scorpions (*Shultz, 2007*; *Pepato, Da Rocha & Dunlop, 2010*; *Garwood et al., 2017*). However, molecular analyses do not agree, placing Opiliones as the sister group to a clade including spiders, Pedipalpi, scorpions, Ricinulei and Xiphosura, although the authors note the impact of long branch attraction (*Sharma et al., 2014*).

### Sexual dimorphism and potential drivers

'Total' body size in Opiliones is typically taken as the length of the dorsal scute, which comprises the dorsal prosomal shield and the first abdominal segments (*Willemart et al., 2009*; *Zatz, 2010*). While this is generally seen as a good metric for quantifying overall body size, some publications report differences in body size based on a number of other characteristics. SSD is reported in numerous harvestman groups. Females in the families Nipponopsalidae, Sclerosomatidae and the genus *Crosbycus* are larger than males, although few males are known in the latter (*Pinto-Da-Rocha, Machado & Giribet, 2007*). The metric used to quantify SD in this instance is not clear, however. Larger body size in females has also been reported in *Longiperna concolor* and *Promitobates ornatus*, based on dorsal scute length (*Zatz, 2010*). Conversely, in Cranaidae and Oncopodidae the carapace is much larger in males than females (*Pinto-Da-Rocha, Machado & Giribet, 2007*). Hence, whilst statistical testing is limited within the Opiliones, this qualitative work suggests the direction of SSD might be variable across the group.

Modification of the tergites, sclerotized upper sections of arthropod segments, is observed in a number of species. In Pettalidae, tergites around the anal region in males possess grooves and ridges that are absent in females; in extreme cases tergites in this region become divided (*Pinto-Da-Rocha, Machado & Giribet, 2007*). Levels of sclerotization can also differ between sexes, as does body patternation (*Pinto-Da-Rocha, Machado & Giribet, 2007*; *Taylor, 2004*). The drivers behind this type of dimorphism are unclear.

Sexual dimorphism and SSD in specific appendages is more strongly supported within Opiliones. In *L. concolor*, for example, the fourth pair of legs displays male-bias SSD in length (*Zatz, 2010*; Fig. 8). Leg length is also bimodal in males of this species: males of the 'major' morph show positive allometry, whilst males of the 'minor' morph are short-legged and display isometry. Thus, 'minor' males that lack the exaggerated features of the 'major' males appear more like females (*Zatz et al., 2011*). Such male dimorphism has been correlated to the presence of intraspecific male fighting, with the fourth leg being used in contests between males of the 'major' morph. 'Minor' males, in contrast, avoid contests and employ a tactic of 'sneaking' into harems in order to steal copulations (*Zatz et al., 2011*). *Willemart et al. (2009)* identify five characters in *N. maximus* that show positive allometry in males, but not in females. All are involved in male–male contests. These include apophyses on the leg four coxae and trochanters, and a dorsal-proximal spine on the femur of the fourth leg, all of which are involved with a

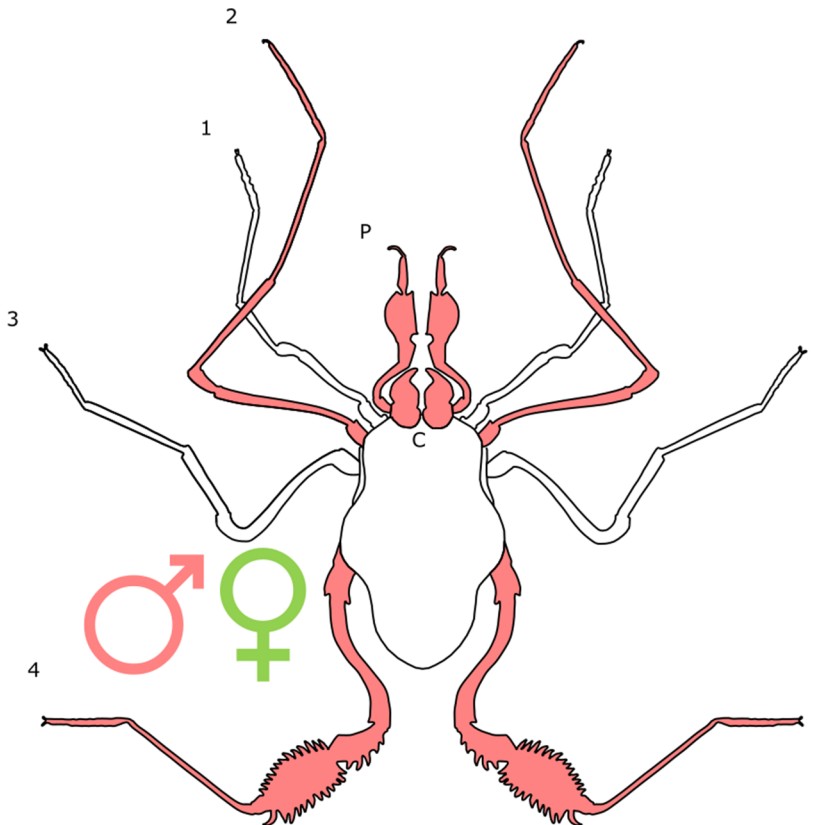

**Figure 8 Patterns of SSD across Opiliones.** See 'Standard Figure Abbreviations' for labelling guide.

phase of fighting termed 'nipping' (*Willemart et al., 2009*). The apophyses take a much simpler form in females (*Willemart et al., 2009*). The curvature and diameter of the males' fourth femur is also characterised by positive allometry, potentially creating an advantage in the 'pushing' phase of contest, in which males use their fourth legs to attempt to move their opponent (*Willemart et al., 2009*).

Similarly, SSD and male dimorphism co-occur in the second leg of *Serracutisoma proximum*. In this species, males of the 'major' morph use the second leg to tap opponents in a ritualised territorial contest (*Buzatto & Machado, 2008*; *Buzatto et al., 2011*), with the winner of such contests either holding, or taking over the contested territory and hypothetically increasing their resource holding potential. Yet field observation, coupled with statistical testing, has revealed no significant difference in second leg length or body size between the winners and losers of territorial contests (*Buzatto & Machado, 2008*). Males with longer second legs do control larger harems, however, but do not hold preferential territories (*Buzatto & Machado, 2008*).

Chemical communication has also been correlated to sex in Opiliones. Tegumental gland openings located on the tarsus of the first, fourth and occasionally third leg, or the femur of leg one, are present in males but not females (*Willemart et al., 2010*; *Proud & Felgenhauer, 2013*; *Da Silva Fernandes & Willemart, 2014*).

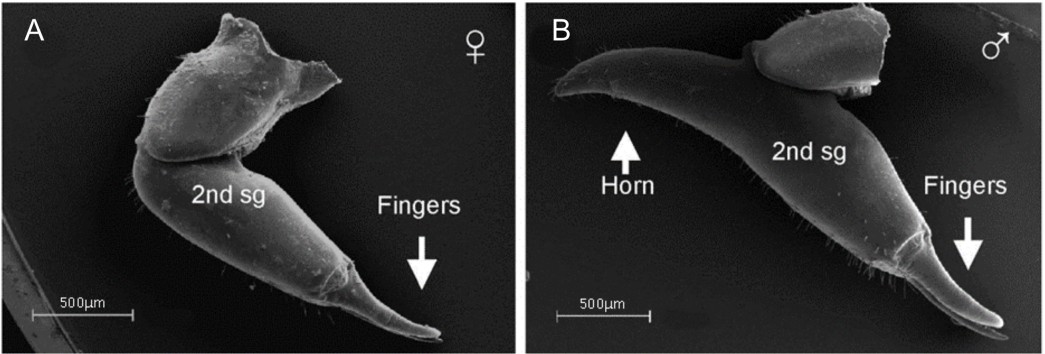

**Figure 9 SEM images showing dimorphism in the chelicerae of *P. opilo*.** The male chelicerae (B) are noted for the presence of a horn used in contest which is absent in the female (A, modified from *Willemart et al., 2006*) © Canadian Science Publishing or its licensors.

Males rub the glandular pores on surfaces, and control the flow of pheromones excreted (*Da Silva Fernandes & Willemart, 2014*; *Murayama & Willemart, 2015*). Meanwhile, female *Dicranopalpus ramosus* possess greater numbers of sensory structures (campaniform and falciform setae) on their tarsi relative to males (*Wijnhoven, 2013*), suggesting females may have an enhanced ability to detect chemical cues left by males. Males do however possess sensilla chaetica, which are also thought to have a chemoreceptive function (*Spicer, 1987*; *Kauri, 1989*; *Willemart et al., 2009*), suggesting that chemical secretions may also play a role in warding off rival males (*Da Silva Fernandes & Willemart, 2014*).

Male-bias SSD is also statistically supported in the pedipalpal length of *Phalangium opilio*, and SD is observed through mechanoreceptors identified solely on the male appendage (*Willemart et al., 2006*). Males of this species fight by pushing against each other and rapidly tapping their pedipalps against the opponent. Pedipalp SSD is thought to determine the strength and frequency of taps (*Willemart et al., 2006*). The appendages are also used to hold the legs of females during copulation, suggesting male pedipalps have adaptations for multiple functions (*Willemart et al., 2006*). Likewise, male-bias SSD is reported in the length of the chelicerae in some families (e.g. Metasarcidae, Cranaidae and Oncopodidae; *Pinto-Da-Rocha, Machado & Giribet, 2007*). In *P. opilio*, male chelicerae also have a horn-like projection protruding upwards in a dorsal direction from the second cheliceral segment (*Willemart et al., 2006*). During contests, males align their chelicerae and push against one another, with the 'horns' providing a surface for the opponent to push against (Fig. 9). Cheliceral horns are also placed over the female dorsum post-copulation, again suggesting multiple functions (*Willemart et al., 2006*). In species characterised by extreme male polymorphism, such as *Pantopsalis cheliferoides*, SD is also reported in chelicerae length, with the smallest male morph typically possessing reduced chelicerae relative to the female (*Painting et al., 2015*).

It is clear that male–male contests and differing mating strategies are a key control on SD in harvestmen, yet recent work has suggested a more fundamental control on

whether males aim to hold territory or favour scramble competition, and thus the potential level of dimorphism observed. Harvestman breeding season length is best predicted by the number of months experiencing favourable climatic conditions, particularly temperature (*Machado et al., 2016*). In climates that consistently experience monthly mean temperatures of over 5 °C along with the requisite amount of precipitation, the breeding season is long and males usually hold reproductive territories. In cooler climates the breeding season is much shorter, and scramble competition is the main mating tactic (*Machado et al., 2016*). The greatly exaggerated contest structures characterised by male-biased SSD are therefore typically only seen in warmer climates (*Machado et al., 2016*).

It should also be noted that SD and male dimorphism often co-occurs in harvestmen, having been attributed to similar selective pressures offset by intralocus sexual and tactical conflict (*Buzatto & Machado, 2014* and references therein). Several studies have differentiated between a 'major' male morph with exaggerated traits and more 'female-like' 'minor' morph. Whilst such studies do not strictly quantify SD, information on male dimorphism can still be informative with regard to alternative mating tactics and the morphological differences between females and males of the 'major' morph. For further information on male dimorphism, we refer readers to *Buzatto & Machado (2014)*, which details male dimorphism in the group.

In conclusion, a male bias in the size of legs, chelicerae and other structures that appear to be related to intrasexual selection are well supported in Opiliones. The common direction of SSD in total body size remains unclear, however, due to ambiguous data with poor statistical support, though it is possible that it varies across the order. Given the large number of studies pointing towards male–male contest as a primary driver in SD in harvestmen it may be expected that, like mammals that exhibit male–male contests, SSD is biased in the direction of males (*Smuts & Smuts, 1993*). However, though contest is clearly a driver for the exaggerated morphologies of 'major' males, comparatively little work appears to have been dedicated to how 'minor' males, where contest is not a factor, differ from females. Identifying a reliable proxy for overall body size and statistically testing SSD should also be a priority.

## Ricinulei

### Description and phylogeny

Ricinulei, or hooded tick spiders, are the least speciose arachnid order comprising only 58 described species (*Prendini, 2011*). Ricinulei appear to inhabit damp tropical environments such as wet leaf litter and caves (*Gertsch, 1971*; *Cokendolpher & Enríquez, 2004*; *Cooke, 1967*; *Tourinho & Azevedo, 2007*). Features of the group include a locking ridge between the prosoma and opisthosoma, and, uniquely, a hood that can cover the mouthparts. No consensus exists on the placement of Ricinulei, which ranges between studies from being the sister group to a clade including Acari and solifuges (*Garwood et al., 2017*), or a clade with Acari (*Shultz, 2007*; *Pepato, Da Rocha & Dunlop, 2010*) to a sister group to Xiphosura (*Sharma et al., 2014*).

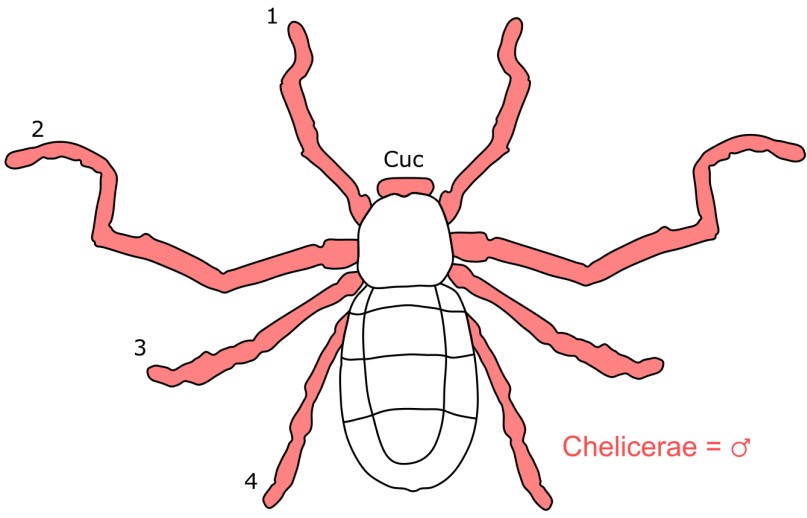

**Figure 10 Patterns of SSD across Ricinulei.** See 'Standard Figure Abbreviations' for labelling guide, Cuc, cucullus.          

### Sexual dimorphism and potential drivers

There is little evidence of SSD in overall body size in Ricinulei, although males of *Pseudocellus pachysoma* have been found to possess a shorter and more granulated carapace than females (*Teruel & Schramm, 2014*). In *Cryptocellus lampeli*, the carapace is broader in females than it is long, whilst the opposite is true in males (*Cooke, 1967*). Dimorphism is present in the third leg across the group, where a copulatory organ is present in males (*Legg, 1976*). The organ derives from modified metatarsal and tarsal podomeres (*Pittard & Mitchell, 1972*). Of particular note is the close correspondence between the margins of the male metatarsal dorsum and a flange on the female's IV coxae (*Legg, 1976*), which become attached during mating (*Legg, 1977*). It is possible that the seemingly co-evolving leg structures could be an example of the 'lock and key' hypothesis (*Masly, 2012*). Adaptations related to copulation in males are thought to be taxonomically informative in the group (*Tuxen, 1974*), but whether these structures contribute to reproductive isolation is yet to be tested. *Cooke & Shadab (1973)* report that the shape of the abdominal sclerites and the number of tubercles can also show significant SD, but do not expand on these statements. SD is also expressed in arrangements of the tubercles found on the pedipalps (*Legg, 1976*).

 Male-biased SSD has also been documented in the legs of Ricinulei (Fig. 10). Based on a small sample size, *Legg (1976)* found all the legs of *Ricinoides hanseni* males to be longer than those of females relative to body length. In the second leg, male femoral diameter can be twice that of conspecific females, and the patella of males is also longer and more curved (*Pittard & Mitchell, 1972*). In *P. pachysoma*, the male first leg is thicker, and has a small conical spur with a coarse granulated texture on its inner surface (*Teruel & Schramm, 2014*). This pattern has been correlated to the complex mating behaviour of Ricinulei, during which males may climb on top of females (*Cooke, 1967*; *Legg, 1976*) and engage in an extended period of 'leg play', where males rub and tap females with legs,

before copulation occurs (*Cooke, 1967*; *Legg, 1977*). This may indicate that female mate choice drives the elongation of male legs.

The retractable 'hood' (cucullus) covering the mouthparts and chelicerae also differs between sexes. It is both wider and longer in male *C. foedus* than females, and is sometimes more reflexed at its edges (*Pittard, 1970*). The cucullus is hypothesised to play a role in mating, the male cucullus acting as a wedge to help unlock the ridge between the prosoma and opisthosoma in females, whilst *Ricinoides afzeli* females use the cucullus to stabilise eggs during transport (*Pittard, 1970*). This suggests that female mate choice and differing reproductive roles may drive cucullus dimorphism. The cucullus also has non-reproductive functions, aiding in capturing prey and holding food during consumption (*Pittard, 1970*) and is therefore also likely under the pressure of natural selection. Male-biased chelicerae SSD has also been reported, but the driver of this dimorphism is unclear (*Legg, 1976*).

To date, most documented instances of SD in Ricinulei are qualitative, and little morphometric data exists to provide statistical support of these conclusions. Future studies would benefit from revisiting previously described collections (*Cooke & Shadab, 1973*) and applying morphometric analyses, allowing the occurrence/extent of SD to be more rigorously quantified.

## Schizomida

### Description and phylogeny

Schizomida, or short-tailed whip scorpions, comprise just over 230 described species (*Reddell & Cokendolpher, 1995*). Most species in the order are primarily tropical in distribution and tend to be found away from bright light, with some species being troglodytes (*Humphreys, Adams & Vine, 1989*). Schizomids have been found in desert environments (*Rowland & Reddell, 1981*) and on the underside of ice and snow covered rocks (*Reddell & Cokendolpher, 1991*), illustrating their climatic range. Morphologically, schizomids resemble whip scorpions, except their prosoma, which is divided into two regions (*Barnes, 1982*), and the lack of eyes. Due to these morphological similarities, schizomids are almost universally thought to be the sister group of Uropygi (*Giribet et al., 2002*; *Shultz, 2007*; *Legg, Sutton & Edgecombe, 2013*; *Garwood & Dunlop, 2014*; *Sharma et al., 2014*).

### Sexual dimorphism and potential drivers

The most consistent sexually dimorphic trait within schizomids is the flagellum (a projection from the terminal opisthosoma), which often varies in shape between sexes. The male flagellum is generally enlarged and bulbous, whereas the female is typically elongate (*Harvey, 2003*). It has been postulated that the flagellum plays a role in sex and species recognition during mating (*Sturm, 1958*, *1973*). Details of courtship and mating are limited to one species (*Surazomus sturmi*), in which the female uses her mouthparts to grip the male flagellum during courtship (*Sturm, 1958*, *1973*). Given that many schizomids have secondarily lost their eyes (*Harvey, 1992*), it is certainly possible that the grasping of the male flagellum plays a role in both sex and species

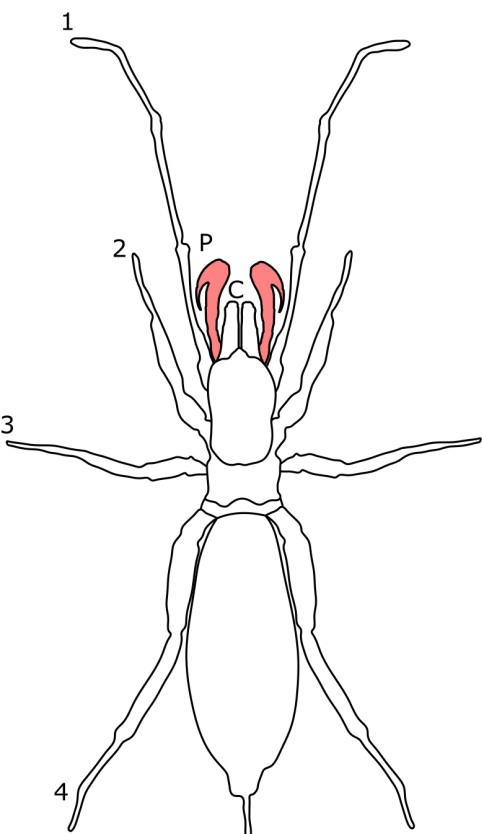

**Figure 11 Patterns of SSD across Schizomida.** See 'Standard Figure Abbreviations' for labelling guide.

recognition during courtship. It has been noted, however, that flagellum dimorphism is absent in other taxa (*Rowland & Reddell, 1980*), with males of the family Protoschizomidae often possessing an elongate flagellum similar to that of females (*Rowland & Reddell, 1979a*). Instead, Protoschizomidae species lacking dimorphism in the flagella tend to show narrowing of the distal body segments in males; elongation is seen in pygidial segments X–XII and/or terminal body segments V–XII (*Rowland & Reddell, 1979a*).

Sexual size dimorphism is also present in the schizomid pedipalp: males of many species have significantly longer pedipalps than conspecific females (*Harvey, 2001*; *Santos, Ferreira & Buzatto, 2013*; *Monjaraz-Ruedas & Francke, 2015*; Fig. 11). In dimorphic species, such as *Rowlandius potiguar*, male pedipalp length is also highly variable relative to prosoma length compared to females (*Santos, Ferreira & Buzatto, 2013*; Fig. 12). This has been attributed to the co-occurrence of male dimorphism, where male morphs with either a long or a short pedipalp are present, the latter having pedipalps similar in shape and size to the female (*Santos, Ferreira & Buzatto, 2013*). Male pedipalpal elongation occurs largely in the femur, patella and tibia (*Rowland & Reddell, 1979a*, *1981*).

In contrast to Opiliones, where male dimorphism has been correlated with male–male fighting (*Buzatto et al., 2011*; *Zatz et al., 2011*), evidence for direct combat in schizomids is lacking. Furthermore, the male pedipalp does not play a direct role in copulation

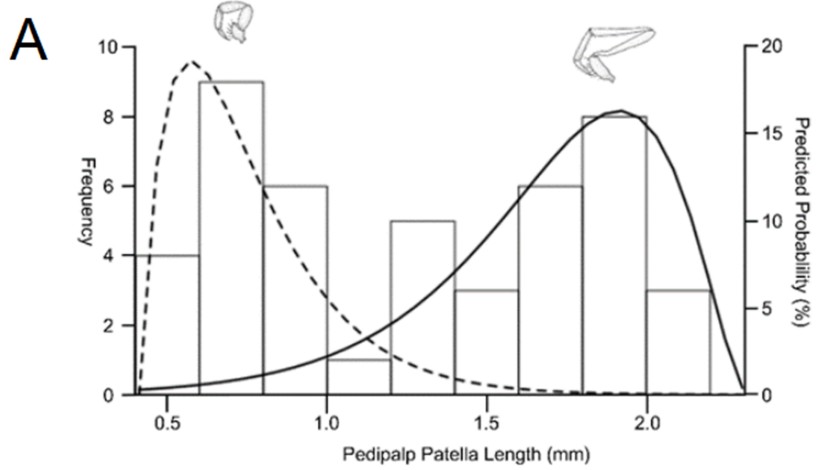

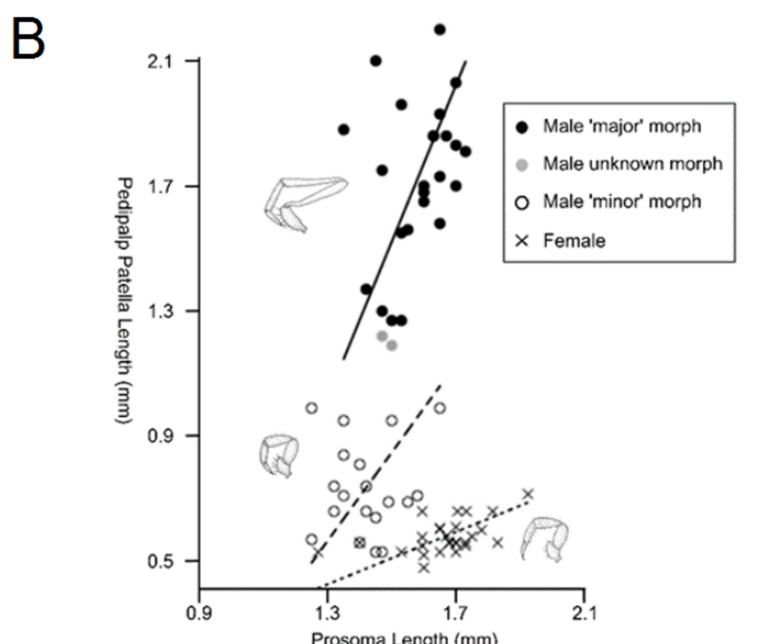

**Figure 12 Patterns of differences in pedipalp lengths denoting both sexual and male dimorphism.**
(A) Frequency histogram of pedipalp patella lengths, (B) relationship between pedipalp patella length and prosoma length for the two male morphs and female (modified from *Santos, Ferreira & Buzatto, 2013*).

(*Sturm, 1958, 1973*). However, observations of the courtship of *Hubbardia pentapeltis* suggest that males stretch out their pedipalps and use them to pick up small twigs before displaying them for females (J.M. Rowland, personal communication from *Santos, Ferreira & Buzatto, 2013*). Further work is required to confirm this within *Rowlandius* and other genera. If this behavioral information is confirmed, it would suggest that female mate choice may be driving dimorphism.

Sexual dimorphism in shape is also present in the schizomid pedipalps. Species of the *Mexicanus* species group (a clade defined by *Rowland, 1975* containing members of

the genus *Schizomus*) show both SD and male dimorphism: some males have a large pedipalp with a tibial spur, which is absent in males with smaller pedipalps and females (*Rowland & Reddell, 1980*).

Sexual dimorphism in schizomids is far from consistent, its presence/absence varying at both a family and genus level (*Rowland & Reddell, 1979a*, *1979b*, *1980*, *1981*). Even within a single species the extent of SD varies in response to the environment. Cave dwelling individuals of *Schizomus mexicanus* are more strongly sexually dimorphic than those of epigean populations, for example (*Rowland & Reddell, 1980*). Whilst compelling evidence has been put forward in support of sexual selection driving schizomid dimorphism (*Santos, Ferreira & Buzatto, 2013*), a paucity of behavioural data limits further understanding. Future research on the potential pressures schizomids face in situ is therefore necessary.

## Scorpiones

### Description and phylogeny

Scorpions are one of the more diverse arachnid orders comprising around 1,750 described species (*Kovarik, 2009*). They have colonised a wide range of terrestrial environments, with a northernmost occurrence of 50°N (*Polis & Sissom, 1990*). Scorpions are unique amongst arachnids in possessing a long metasoma (tail) terminating in a venomous sting. Significant uncertainty exists regarding the placement of the group within the arachnid phylogeny. Recent morphological analyses have suggested they could be the sister group of harvestmen (*Shultz, 2007*), the sister group to a clade of solifuges and pseudoscopions (*Wheeler & Hayashi, 1998*; *Giribet et al., 2002*), the sister group to Opiliones and pseudoscorpions (*Garwood et al., 2017*) or the sister group to pseudoscorpions (*Pepato, Da Rocha & Dunlop, 2010*). Molecular phylogenies variously place the order as closest to Ricinulei and Pedipalpi (*Sharma et al., 2014*), or as the sister group to Pseudoscorpions, solifuges and harvestmen (*Giribet et al., 2002*). One placement that has gained recent traction is Arachnopulmonata, a clade that includes scorpions and pantetrapulmonata (spiders and pedipalpi). This clade has been recovered from molecular studies (*Sharma et al., 2014*) and the groups within the clade seems to have morphological similarities in their vascular systems (*Klußmann-Fricke & Wirkner, 2016*; see also *Giribet, 2018*).

### Sexual dimorphism and potential drivers

Sexual size dimorphism in scorpions is relatively consistent across the group (Fig. 13). Females typically have a larger carapace than males, which is thought to be a reliable indicator of overall body size (*Koch, 1977*; *Sánchez-Quirós, Arévalo & Barrantes, 2012*). Nevertheless, the extent of SSD can vary considerably. Australo-Papuan scorpions are characterised by extreme SSD, with the carapace of females on average 40% longer than that of males. In contrast, some species show less than 1% difference in carapace length between sexes (*Koch, 1977*; *Polis & Sissom, 1990*). Reverse SSD is also occasionally observed in some scorpion clades. For example, male *Liocheles australaisae* carapace length is on average 28% greater than that of females (*Koch, 1977*). Female-biased SSD

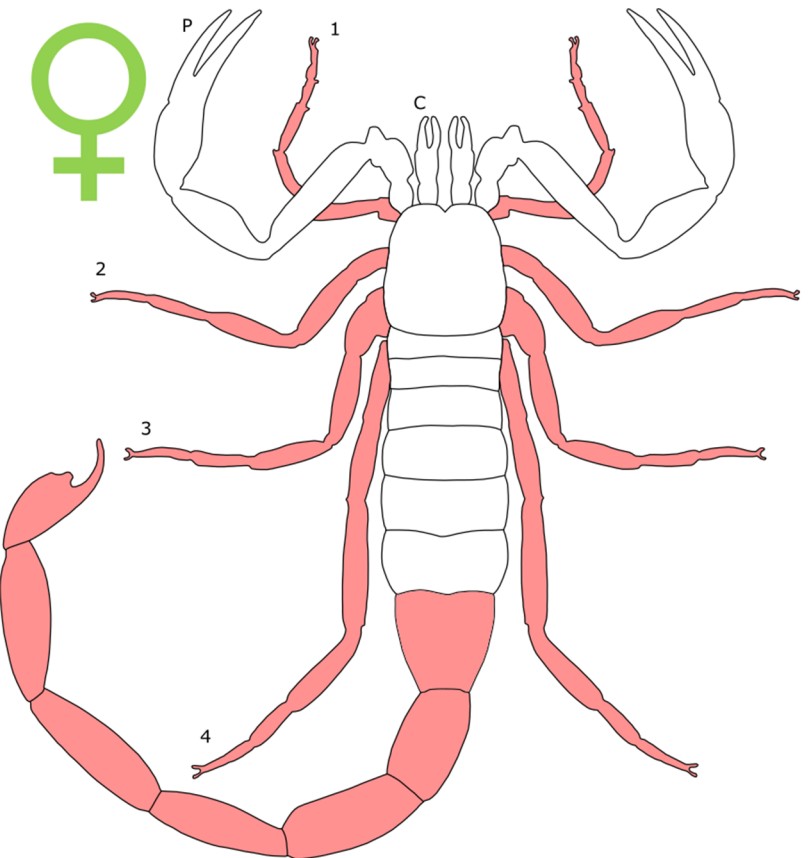

**Figure 13 Patterns of SSD across Scorpiones.** See 'Standard Figure Abbreviations' for labelling guide.

appears to be related to fecundity selection, with clutch size being strongly correlated with maternal body size (*Outeda-Jorge, Mello & Pinto-Da-Rocha, 2009*).

Scorpion SSD has also been reported based on total body length *inclusive* of tail. *Kjellesvig-Waering (1966)* found males of *Tityus tritatis* to be longer in overall body length than females. We note that this length metric is likely a poor proxy for total body size, as the metasoma of male scorpions (segments comprising the tail exclusive of the telson) is often elongated (*Koch, 1977*; *Carlson, McGinley & Rowe, 2014*; *Fox, Cooper & Hayes, 2015*); a trait most marked in the genera *Centruoides, Hadogenes, Isometrus* and *Hemiscorpius* (*Polis, 1990*). This elongation is achieved by lengthening of existing metasomal segments relative to females (*Carlson, McGinley & Rowe, 2014*), rather than the addition of segments. As such, total body length performs worse than carapace length as a predictor for body mass, due to the confounding factor of SSD in the tail. The telson itself is not sexually dimorphic in the majority of species, but there are some exceptions (*Polis & Sissom, 1990*). In *Heterometrus laoticus* the telson is longer in males (*Booncham et al., 2007*). Other structural modifications can be found in males of *Anuroctonus, Chaerilus* and *Hemiscorpius* (*Polis & Sissom, 1990*; *Lourenço & Duhem, 2010*) and there is even some evidence of dimorphism in venom glands in scorpions that exhibit sexual stinging (*Sentenská et al., 2017*).

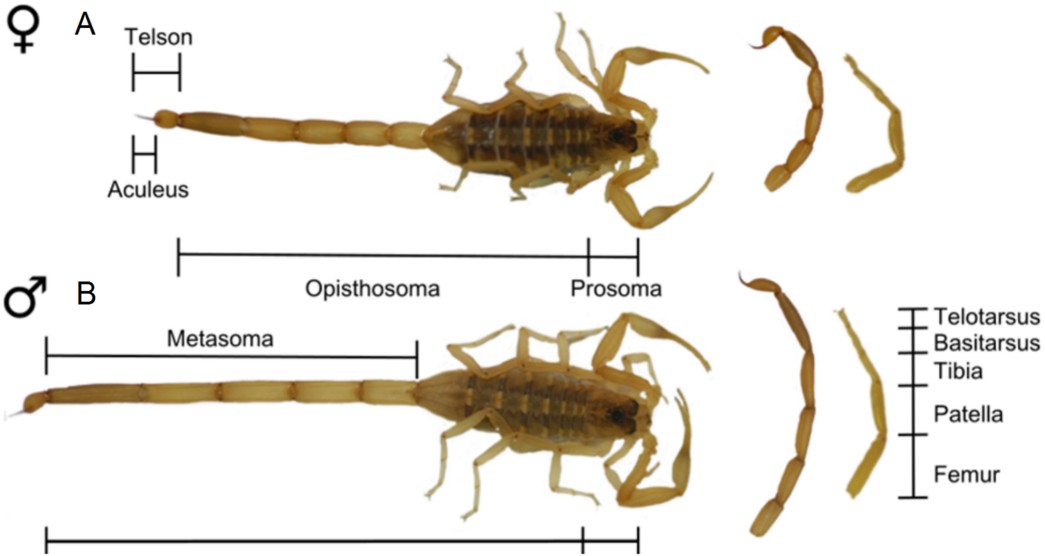

**Figure 14 Sexually dimorphic body plan of *Centruroides vittatus*.** Differences between the female (A) and male (B) body plan in *Centruroides vittatus*, note the longer metasoma and legs in the male.

The extent to which tail SSD is reflected in behavioural differences between male and female scorpions remains unclear. Lengthening of the male metasoma has no impact on either sprinting performance (by acting as a counterweight) or sting performance, defined as the number of discrete stings when antagonised within a given time period (*Carlson, McGinley & Rowe, 2014*). It may be that the increased length of the male metasoma is related to 'sexual stinging', in which males sting their prospective mates (often in the arthrodial membrane adjacent to the pedipalpal tibia) to stun the female and facilitate mating (*Angermann, 1955*, *1957*; *Francke, 1979*; *Tallarovic, Melville & Brownell, 2000*). The male metasoma may also be used to 'club' or rub the female during mating (*Alexander, 1959*; *Polis & Farley, 1979a*).

The limbs of scorpions are also characterised by SSD, with male *Centruroides vittatus* possessing significantly longer legs relative to total body size than females (Fig. 14). This translates to a 30% sprint speed increase over females of the same body size (*Carlson, McGinley & Rowe, 2014*). Limb elongation has therefore been linked to the documented male 'flight' vs. female 'fight' response to predation (*Carlson, McGinley & Rowe, 2014*). Similar locomotory benefits could potentially also apply to males seeking out sedentary females prior to mating. Finally, longer legs could also aid 'leg play' during mating (*Polis, 1990*).

In common with other arachnids (e.g. Schizomida and Amblypygi), marked dimorphism is present in the pedipalps, which carry claws (chelae) in scorpions. Chelae in males are often described as elongate or gracile compared to females, although the opposite is observed in some genera (e.g. *Buthus*, *Scorpio* and some *Titus*; *Polis, 1990*). The degree to which male chelae really are larger than females after controlling for body size remains a point of contention, however. Whilst both the fixed and movable fingers of male chelae are longer and wider than females in absolute terms across

numerous species (e.g. *Caraboctonus keyserlingi*, *Pandinus imperator* and *Diplocentrus* sp.; *Carrera, Mattoni & Peretti, 2009*), no analyses normalise against body length. This largely reflects the above difficulties (as discussed in above) in identifying a reliable reference character for overall body size in Scorpions (*Fox, Cooper & Hayes, 2015*). In contrast, dimorphism in chelae shape is more strongly supported. In a number of species, the movable finger of females is more curved than that of the males (*Carrera, Mattoni & Peretti, 2009*), and dentition (processes on the inside surface of the chelae) differs between sexes in the family Buthidae (*Maury, 1975*). Pedipalp dimorphism has previously been hypothesized to play a role in mating. During courtship, many scorpions act in a 'courtship dance' involving the male and female grasping chelae prior to mating (*Alexander, 1959*; *Polis & Farley, 1979a*). Dimorphism in pedipalpal chelae dentition, in particular, is thought to aid the male's grip of the female during mating (*Maury, 1975*).

Sex differences in mode of life have also been proposed as potential drivers of dimorphism in the scorpion pedipalpal chelae and chelicerae (*Carrera, Mattoni & Peretti, 2009*). Males are more active during the mating season than females (*Polis & Sissom, 1990*) and excavate burrows more frequently than females (*Carrera, Mattoni & Peretti, 2009*). In contrast, females build specialised burrows for maternal care (*Polis, 1990*). Interspecific morphological differences associated with burrowing are common (*Polis, 1990*; *Prendini, 2001*), but burrowing has yet to be systematically investigated as a driver behind SD in scorpions.

Finally, marked SD is also observed in the pectines, a ventral wing-shaped structure with numerous teeth, used a sensory organ. Females have smaller pectines than males, and the angle between the two wings is greater (*Polis, 1990*). In an ontogenetic study of *Paruroctonus mesaensis*, male pectines grew at a much faster rate when the animal reached sexual maturity, potentially indicating the organ may be subject to sexual selection (*Polis & Farley, 1979b*). Multiple authors have also found statistically significant differences in pectine length between species (*Booncham et al., 2007*; *Fox, Cooper & Hayes, 2015*). Pectines function as both mechano- and chemoreceptors. It has been hypothesised that males use their larger structures to track chemical trails left by females, and thus find mates (*Melville, 2000*). Several authors have also suggested that males have more pectinal teeth than females (*Alexander, 1959*; *Williams, 1980*; *Mattoni, 2005*).

In summary, SSD is less extreme in scorpions than many other arachnid groups, yet several anatomical regions do reliably exhibit sex differences. On average, females are larger in total body size, whilst males possess longer legs, elongate and gracile chelae, a slender metasoma and enlarged pectines. Reverse SSD is present in the chelae and metasoma in some groups (*Polis & Sissom, 1990*). Future research should aim to map the phylogenetic distribution of such traits in order to better understand how life history and habitat use may result in differential selection operating on males and females.

## Solifugae

### Description and phylogeny

Solifuges, known as camel spiders or sun spiders, comprise approximately 1,000 species (*Punzo, 1998a*). The order is largely limited to arid environments, although some species

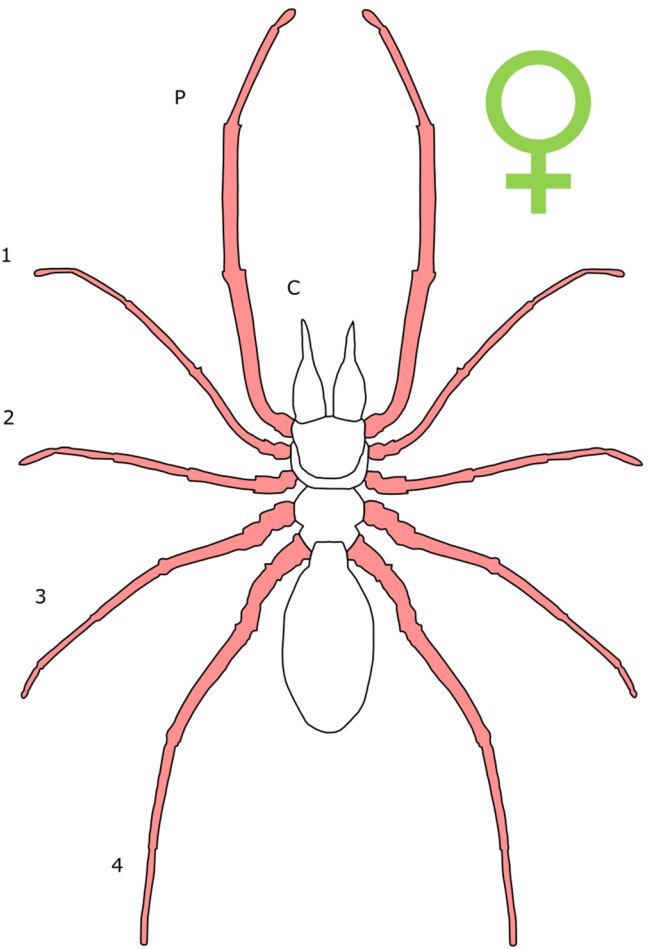

**Figure 15 Patterns of SSD across Solifugae.** See 'Standard Figure Abbreviations' for labelling guide.

are found in rainforests and their margins (*Harvey, 2003*). The occurrence of sensory racquet organs on the ventral surface of the coxae on leg IV differentiate Solifugae from other arachnids. Other notable morphological features include enlarged chelicerae, elongate leg patellae relative to other arachnids and the presence of trachea instead of book lungs (*Harvey, 2003*). There is some debate over their phylogenetic position within arachnids. Some studies report solifuges as the sister group to pseudoscorpions (*Shultz, 2007*; *Giribet et al., 2002*) while others place them in a clade with Acariformes (*Pepato, Da Rocha & Dunlop, 2010*, *Garwood et al., 2017*). Recent molecular work has placed solifuges as the sister group to a clade including Xiphosura, Ricinulei, Scorpiones, Pedipalpi, Araneae and Opiliones (*Sharma et al., 2014*).

***Sexual dimorphism and potential drivers***

Body length SSD is present in solifuges. Males are typically slightly smaller in body size, more slender in form, and have longer limbs than females (*Punzo, 1998b*; *Peretti & Willemart, 2007*; Fig. 15). Female-biased SSD likely relates to a fecundity advantage, with body size tightly correlating to clutch size in *Eremobates marathoni* (*Punzo, 1998a*).

It has been suggested that the longer legs of males in Solifugae could relate to extended mate searches or use in mating (*Wharton, 1986*). Racquet organs are also larger in males (*Peretti & Willemart, 2007*), and their hypothesized function as chemoreceptors may increase male capacity to detect pheromones and aid mate search (*Punzo, 1998a*). The fact that male pedipalps are used to 'massage' the female during mating (*Heymons, 1902*; *Junqua, 1962*) may also explain why all male limbs are elongated relative to overall body size.

Amongst arachnids, solifuges are best recognised by their large chelicerae. Numerous studies report SD in the chelicerae (see Supplementary Material), yet often fail to distinguish the effects of *shape* and *size* dimorphism from one another. Indeed, a commonly reported metric of solifuge chelicerae is their aspect ratio, with male chelicerae characterised by a greater length:width ratio than those of females (*Punzo, 1998a*; *Peretti & Willemart, 2007*). Whilst aspect ratio can itself be an important metric, often affecting function (*Kruyt et al., 2014*; *Yeh & Alexeev, 2016*), the degree to which the 'slender' chelicerae of males are also dimorphic in total size is yet to be addressed in the literature. Calculations based on mean values presented by *Punzo (1998a)* do suggest female-biased dimorphism in cheliceral length and width, however. Quantifying the presence of SSD in chelicerae is further complicated by the lack of a reliable metric for total body size. Body length has been considered problematic, as the size of the abdomen is known to increase post-feeding (*Brookhart & Muma, 1981*; *Wharton, 1986*). Elsewhere, the CP index, the combined length of the chelicerae and propeltidium (the prosomal dorsal shield in solifuges) has been preferred as a metric of solifuge total body size (*Bird, 2015*), further confusing the picture with regards to chelicerae length and overall SSD.

Dimorphism in solifuge chelicerae shape and dentition (projections from the chelicerae) is more widely accepted. Male chelicerae are straighter (*Hrušková-Martišová, Pekár & Bilde, 2010*), the fixed finger is less curved and the manus (a broad proximal section of the paturon which contains the cheliceral muscles) is more gracile, that is, narrower than in females (*Bird, 2015*). The dentition of adult male chelicerae is also reduced in projection size (*Bird, 2015*). This is not universally true, however—though not quantified, there appears to be little to no difference in the size of the primary and secondary teeth between sexes in *Solpugiba lineata* and some species of *Hemiblossia* (*Bird, 2015*). Both are known to be termitophagous, thus *Bird (2015)* has hypothesised that solifuge cheliceral dimorphism is linked to feeding behaviour. Males are known to feed less often than females (*Junqua, 1962*; *Wharton, 1986*), and male chelicerae show less dental wear (*Fitcher, 1940*). Sex differences in dietary preference have also been observed under laboratory conditions, with female *Gulvia dorsalis* feeding on highly sclerotized beetles, which are refused by males (*Hrušková-Martišová, Pekár & Bilde, 2010*). The increased depth of the manus in female chelicerae may therefore facilitate an increase in muscle volume and enhanced bite force and feeding efficiency (*Bird, 2015*). Such a pattern has previously been found interspecifically: species characterised by chelicerae that are more robust are capable of delivering a stronger bite force (*Van Der Meijden et al., 2012*).

Alternatively, dimorphism in solifuge chelicerae may arise from their function during mating (*Van Der Meijden et al., 2012*). Male *Galeodes caspius* use their chelicerae to insert

spermatophores into the genital opening of the female (*Hrušková-Martišová, Pekár & Bilde, 2010*), often inserting the fixed finger or occasionally the whole chelicera into the genital opening (*Amitai, Levy & Shulov, 1962*; *Bird, 2015*). After sperm transfer, the male may start a 'chewing' action; the precise reason for this is unknown but is hypothesised to help force sperm into a storage area and/or break up the spermatophore (*Muma, 1966*). The straighter shape of the male chelicerae may assist with spermatophore insertion (*Hrušková-Martišová, Pekár & Bilde, 2010*), whilst reduced dentition could minimise damage during genital chewing (*Bird, 2015*). Sexually dimorphic setae are also present on the base of the chelicerae, In *Oltacola chacoensis*, for instance, these are less numerous in males, but larger and harder (*Peretti & Willemart, 2007*). During mating, setae are pressed up against the perigenital region of the female, indicating a potential role during mating (*Peretti & Willemart, 2007*).

Sexual dimorphism is also present in the solifuge flagellum, an elongate structure protruding from the fixed finger of the chelicerae. The flagellum occurs only in male solifugae (*Punzo, 1998a*). There is considerable interspecific variation in both the form of the flagellum (*Lawrence, 1954*; *Punzo, 1998b*) and in its articulation: it is fixed in some species and movable in others (*Punzo, 1998b*). *Lamoral (1975)* suggested multiple potential functions for the flagellum, including as a mechanoreceptor and being involved the storage and emission of exocrine secretions. Flagella may also play a role in mating, being used by male *O. chacoensis* to carry spermatophores (*Peretti & Willemart, 2007*), and being inserted into the genital opening during sperm transfer by male *Metasolpuga picta* (*Wharton, 1986*).

To summarise, SSD is present to some degree in total body size and may be present in chelicerae of solifuges, though shape dimorphism is better accepted. More work is required to determine the relative importance of mating and feeding on cheliceral morphology. *Bird (2015)* advocates a geometric morphometrics approach to quantifying the morphology of chelicerae, and we concur that such a study including males and females from multiple, phylogenetically disparate species would be an important advance in the field. Furthermore, life history information pertaining to Solifugae is limited to a small number of species; mating, in particular, has only been studied in three families (*Hrušková-Martišová, Pekár & Bilde, 2010*). Focusing basic research onto lesser-studied groups may illuminate further trends in SD across the order.

## Uropygi

### Description and phylogeny

Uropygi, known as whip scorpions or vinegaroons, are represented by 110 extant species (*Zhang, 2011*). The group is found in habitats limited to tropical and subtropical areas, preferring damp and humid conditions, although *Mastigoproctus giganteus* is found in arid environments in the southern United States (*Kern & Mitchell, 2011*). As their common name suggests, uropygid morphology bears some resemblance to that of scorpions, with palpal claws and a segmented opisthosoma. However, whip scorpion anatomy differs from that of scorpions in having a segmented terminal flagellum instead of a stinging tail. Furthermore, whip scorpions spray a noxious mixture primarily composed

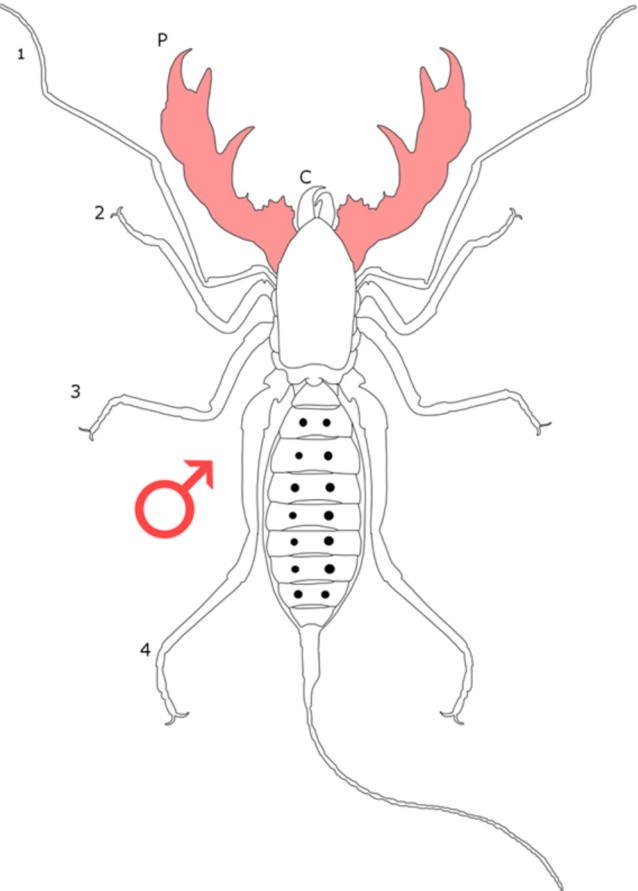

**Figure 16 Patterns of SSD across Uropygi.** See 'Standard Figure Abbreviations' for labelling guide.

of acetic acid from glands located near the pygidium as a means of defence (*Schmidt et al., 2000*). There is consensus in the phylogenetic position of Uropygi: they are widely regarded as the sister group to Schizomida, together forming Thelyphonida, and being united with the Amblypygi to form the clade Pedipalpi (*Giribet et al., 2002*; *Shultz, 2007*; *Sharma et al., 2014*; *Garwood et al., 2017*).

### Sexual dimorphism and potential drivers

Sexual size dimorphism has been reported in whip scorpions, with males having a larger prosomal scutum, the dorsal sclerotized prosomal plate (seen as a good indicator of body size) than females (*Weygoldt, 1988*; Fig. 16). Other minor structural modifications can also be seen in the opistisoma and first leg of females (*Huff & Prendini, 2009*). In the pedipalps, SD is present beyond the fourth nymphal phase, which is the final nymphal stage before maturity. There is an increased positive allometric relationship in the length of the palpal femur and patella when regressed against carapace length in adult male of the species *Mastigoproctus giganteus*, that is, unseen in females (*Weygoldt, 1971*). SSD in the pedipalps is also seen in the genera *Thelyphonellus* and *Typopelti*, and to a lesser degree *Thelyphonus* (*Weygoldt, 1988*). Male pedipalps have also been described as

'stronger'in these genera (*Weygoldt, 1988*), but there are no biomechanical analyses to support this statement. Minor differences in structure between the male and female pedipalps are also present. For example, the third spine on the female trochanter of *Thelyphonus indicus* is much longer relative to other pedipalpal spines (*Rajashekhar & Bali, 1982*), and the patella apophyses are thicker relative to length in females (*Rajashekhar & Bali, 1982*).

The tibial apophysis of the pedipalp in whip scorpions is also dimorphic, though not in every group (*Gravely, 1916*). Where present, dimorphism is expressed through a larger tibial apophysis in males; this results, in males possessing a broader area on the tibia termed a 'palm', which is a consistent feature across Uropygi (*Gravely, 1916*; *Weygoldt, 1971*, *1972*; *Rajashekhar & Bali, 1982*). The tibial apophysis has a wide range of male morphologies across the group, ranging from a small projection to a suite of highly modified curved structures (*Gravely, 1916*). Similarly, the tarsus is characterised by sexually dimorphic projections in some species, with male *T. indicus* (*Rajashekhar & Bali, 1982*) and *M. gigantus* (*Weygoldt, 1971*) bearing a spine close to the tip of the fixed finger of the pedipalpal claw, not present in females.

The sexually dimorphic pedipalps of Thelyphonidae are hypothesized to play a role in male–male contest over prospective females (*Watari & Komine, 2016*). Fighting includes a phase of grappling, where males face each other and fight using their pedipalps, and a tackling phase, during which males try to overturn their opponent using the pedipalps (*Watari & Komine, 2016*). Numerous publications report that males also use the pedipalps in mating, typically grabbing the first legs of the female with the pedipalps and manipulating her until they are face-to-face (*Weygoldt, 1971*, *1972*).

Further work is needed to determine the underlying drivers of SD in the Uropygi. As many species are known from only a small number of individuals (*Gravely, 1916*; *Huff & Prendini, 2009*), a concerted collecting effort will be required before any broad scale patterns in SSD may be distinguished in whip scorpions.

## DISCUSSION

### Trends in SD across Arachnida

When SD is considered across Arachnida as a whole, general trends become apparent (Table 1). The lack of current consensus regarding phylogenetic relationships between arachnid orders precludes us from deriving the ancestral condition of dimorphism, with only Arachnopulmonata (containing Scorpiones, Araneae, Amblypygi, Schizomida and Uropygi; Fig. 17) and its internal relationships being consistently recovered (*Giribet, 2018*). However, a current consensus phylogeny is included to allow readers to gain an insight into the distribution of SD across the group (Fig. 17).

Firstly, though generally not as pronounced as in Araneae, female-biased SSD in overall body size is present across much of Arachnida: female-biased SSD has also been reported in mites, amblypygids, harvestmen, pseudoscorpions, scorpions and solifuges. Whilst some species are known to subvert the general trend, we note that there is no evidence of male-biased SSD being dominant across an order.
**Table 1  Patterns of SSD across arachnid orders.**

|  | Acari | Amblypygi | Aranaea | Palpigradi | Pseudoscorpiones | Opiliones | Ricinulei | Schizomida | Scorpiones | Solifugae | Uropygi |
|---|---|---|---|---|---|---|---|---|---|---|---|
| Overall body | (♀) | ♀ | ♀(♂) |  | ♀(♂) | ♀♂ |  |  | ♀(♂) | ♀ | ♂ |
| Legs |  | ♂* | ♂ |  |  | ♂ | ♂ |  | ♂ | ♂ |  |
| Chelicerae | ♂ |  | ♀♂ |  |  | ♂ | ♂ |  |  | (♀) |  |
| Pedipalps | ♂ | ♂ |  |  | ♀♂ | ♂ |  | ♂ |  | ♂ | ♂ |

Notes:
   ♂/red = male biased, ♀/green = female-biased, symbols in brackets indicate rare reversals, * indicates antenniform legs.

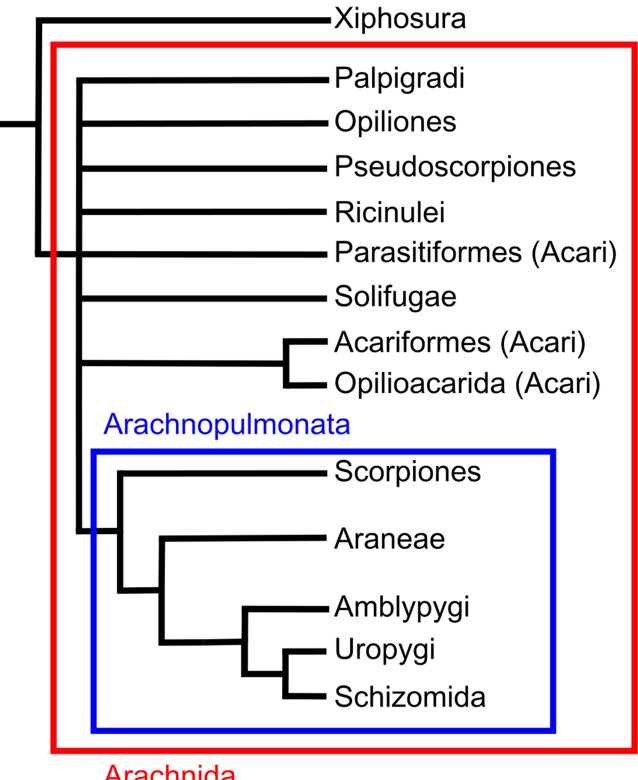

**Figure 17  A broad consensus arachnid phylogeny encompassing a range of recent studies.** A broad consensus arachnid phylogeny encompassing a range of recent studies (modified from *Giribet, 2018*).

    Secondly, SSD in leg length relative to body size typically favours males, occurring in scorpions, solifuges, spiders, ricinuleids and harvestmen. This trait is seemingly driven by behavioural factors, although the precise mechanism differs between groups (see below). Additionally, the majority of arachnid orders exhibit dimorphism in either size or shape of the pedipalps. When present, SSD in the pedipalps typically favours males, which often possess additional spurs or other accessories to the appendage. In the most extreme examples, spiders have modified their pedipalps to transfer spermatophores directly. However, in the majority of cases, the pedipalp does not play a direct role in sperm transfer and is instead involved in female mate choice or intraspecific male contest.

    Sexual size dimorphism in chelicerae is also observed in a number of arachnid orders (Acari, Araneae, Opiliones and Solifugae), though the direction of dimorphism can

differ. When dimorphism is male-biased, the chelicerae tend to be under the influence of sexual selection. For example, Opiliones chelicerae are used in male–male contest (*Willemart et al., 2006*), spider chelicerae are thought to be used for intersexual agonistic displays (*Faber, 1983*) and nuptial gift giving (*Costa-Schmidt & Araújo, 2008*). Female-biased dimorphism, on the other hand, appears to be related to increased feeding due to the high energetic costs of producing eggs. Female biased intersexual difference in the number of prey captured has been empirically demonstrated in spiders that exhibit female-biased cheliceral SSD (*Walker & Rypstra, 2002*). Differences in cheliceral wear patterns suggest this is also the case in solifuge (*Fitcher, 1940*).

Several orders also show male-bias in the number of sensory structures (Amblypygi, Solifugae and Scorpiones). In solifuges and scorpions, the co-occurrence of larger sensory structures and longer leg length (*Melville, 2000*; *Peretti & Willemart, 2007*; *Punzo, 1998b*) may be tied to the selective pressures of mate searching (*Punzo, 1998a*; *Melville, 2000*). In Opiliones, male and females have different sensory anatomy (*Wijnhoven, 2013*) though there is no clear indication as to whether one sex has increased sensory capabilities relative to the other.

## Selective pressures for SD in Arachnida

### *Weapons and ornaments*

When sexually dimorphic structures appear better developed in males, they are often found to play a role in male–male contests or male–female courtship. The degree to which these intra- or intersexual selection pressures are most prevalent has yet to be discussed for Arachnida as a whole, however. Here, we find evidence for male–male contests driving the evolution of sexually dimorphic structures in Acari, Amblypygi, Araneae, Opiliones, Pseudoscorpiones and Uropygi. In mites, male *C. berlesei* use enlarged third legs to kill rival males (*Radwan, 1993*), whilst male amblypygids 'fence' each other using their sexually dimorphic antenniform 'whip' legs (*Weygoldt, 2000*). The hyperallometric chelicerae of male Areneae are known to be used in male–male contests (*Funke & Huber, 2005*), and the enlarged fourth leg of male Opiliones is used in contests between males of the 'major' morph (*Zatz et al., 2011*). Finally, the sexually dimorphic pedipalps of Pseudoscorpiones (*Weygoldt, 1966*; *Thomas & Zeh, 1984*) and Uropygi (*Watari & Komine, 2016*) are involved in grappling during male–male aggression.

Yet in the instances outlined above, the male-biased sexually dimorphic structures have also been found to function during courtship and mating. Elaborations on the enlarged third legs of mites may assist males in aligning with the female spermaduct opening (*Gaud & Atyeo, 1979*), and the sexually dimorphic antenniform 'whips' of amblypygids are also used to display to and rub females prior to mating (*Weygoldt, 2000*). The enlarged chelicerae of some male spiders are thought to play a role in courtship displays (*Faber, 1983*), whilst the pedipalps of pseudoscorpions are also involved in a ritualised dance prior to mating (*Weygoldt, 1966*). There are several instances therefore of *both* intra- and intersexual selection pressures acting on a given sexually dimorphic structure.

Arguably, however, examples of courtship and female choice driving the evolution of sexually dimorphic structures are even more widespread. Of those groups considered in the present study, evidence of intersexual selection driving SSD is lacking for only Uropygi. In addition to the examples listed above, the cheliceral horns of Opiliones are placed on the female dorsum after copulation (*Willemart et al., 2006*), and the longer male legs of Ricinulei are engaged in 'leg play' prior to mating (*Cooke, 1967*; *Legg, 1977*). In schizomids, the female chelicerae grip the male flagellum during mating (*Sturm, 1958*, *1973*), whereas the dimorphic chelicerae of solifuge are used by the male to grip the female and transfer spermataphores (*Peretti & Willemart, 2007*). The dimorphic pedipalp of scorpions has also been hypothesised to play a role in the 'courtship dance', as males and females grasp chelae prior to mating (*Alexander, 1959*; *Polis & Farley, 1979a*). Indeed, in four orders (Ricinulei, Schizomida, Solifugae and Scorpiones), courtship and mating appear to be the primary drivers of male-biased SSD in the appendages.

### Scramble competition

The scramble competition hypothesis posits that the most mobile males within a population will reach and copulate with a greater number of females (*Ghiselin, 1974*). Male traits conferring an advantage in locating a receptive female, such as sensory and locomotor adaptations, may therefore become sexually dimorphic under the selective pressure of scramble competition (*Andersson, 1994*). This is well-supported in the case of Araneae, with decreased male body size and increased leg length in spiders being linked to improved climbing ability (*Moya-Laraño, Halaj & Wise, 2002*), bridging ability (i.e. walking upside-down on silk bridges; *Corcobado et al., 2010*) and locomotor speed (*Grossi & Canals, 2015*). Here, we also identify instances of male-biased SSD in leg length in Acari, Scorpiones, Solifugae, Ricinulei and Opiliones, and reduced total body size in male Acari, Amblypygi, Pseudoscorpions, Scorpiones and Solifugae. Within scorpions, decreased body mass and elongate legs have been correlated to increased sprint speed in male *C. vitttus* (*Carlson, McGinley & Rowe, 2014*), and the increased size of pectines (sensory organs) in males has been hypothesised to play a role in mate searching (*Melville, 2000*). Elsewhere, smaller body size and increased leg length in male Solifugae may also be related to mate searching (*Peretti & Willemart, 2007*), with male *M. picta* typically covering much greater straight-line distances than females (*Wharton, 1986*). The chemosensing racquet organs of male solifuges are also enlarged (*Peretti & Willemart, 2007*). The case for scramble competition driving some aspects of SD in both Scorpiones and Solifugae is therefore convincing. Yet within Ricinulei and Opiliones, male-biased SSD in leg length appears better explained by their role in mating (*Legg, 1977*) and male–male contests (*Willemart et al., 2009*; *Buzatto et al., 2014*), respectively. As will be discussed below, further experimental work focusing on the biomechanical and physiological implications of body size and leg length dimorphism would be particularly insightful in this respect.

### Fecundity selection

Fecundity selection is a well-documented driver of female-bias body size dimorphism within Araneae (*Head, 1995*; *Coddington, Hormiga & Scharff, 1997*). In females of the wolf

spider *D. merlini* the disproportionately large opisthosoma of females has been correlated to egg production and storage, for example (*Fernández-Montraveta & Marugán-Lobón, 2017*). Under laboratory conditions, female body mass in the ant-eating spider *Z. jozefienae* has been found to tightly correlate to number of eggs present within the egg sack (*Pekár, Martišovà & Bilde, 2011*). More broadly across Araneae, body size dimorphism has been explained by female size increase via fecundity selection (*Prenter, Elwood & Montgomery, 1999*; *Huber, 2005*). Yet despite this wealth of data pertaining to Araneae, relatively little is known of the role of fecundity selection across the smaller arachnid orders. Within scorpions, the carapace length of females is correlated to increased litter size (*Outeda-Jorge, Mello & Pinto-Da-Rocha, 2009*), and female-biased dimorphism in prosoma length has therefore been taken as evidence of fecundity selection (*Fox, Cooper & Hayes, 2015*); similar patterns can also be seen in solifuges (*Punzo, 1998a*). Beyond this, female-biased SSD has been identified in other metrics of 'total body size' in harvestmen (*Pinto-Da-Rocha, Machado & Giribet, 2007*; *Zatz, 2010*), pseudoscorpions (*Zeh, 1987a*) and amblypygids (*McArthur et al., 2018*). Whilst the degree to which such dimensions correspond to potential fecundity in these groups has remained largely unexplored. At least in one species of amblypygid, for instance, female carapace size does appear to be correlated to brood size (*Armas, 2005*).

### Niche partitioning

Males and females may also diverge in their energetic requirements due to their different reproductive or social roles, resulting in different trait optima between the sexes (*Slatkin, 1984*). Here, we highlight examples of niche partitioning within Acari and Araneae, although unequivocal examples are limited across Arachnida. Due to the increased energetic demands of reproduction, female ant-eating spiders (*Z. jozefienae*) have been found to consume larger prey items using their enlarged chelicerae compared to males (*Pekár, Martišovà & Bilde, 2011*). In such instances, fecundity selection (as discussed above) can be thought of as driving niche partitioning. The increased reproductive output of females can necessitate habitat or dietary divergence, resulting in morphological dimorphism *beyond* that of total body size. Trophic dimorphism has also been reported in the nymphal stages of Kiwi bird feather mite *Kiwialges palametrichus* (*Gaud & Atyeo, 1996*), with males and females diverging in their preferred microhabitat in and around the feather. In this instance, however, SD and niche partitioning is also compounded by ontogenetic nymphal stages. Hence, whilst there is some evidence that niche partitioning promotes SD in arachnids, it does not currently appear to be a major driving force. The relative lack of examples of niche partitioning (in comparison to male contests, for example) may partly reflect the paucity of information relating to the discrete dietary and habitat preferences of each sex, however. In some instances, our understanding of the differing morphology between sexes far exceeds that of their potential dietary and habitat niches.

## CONCLUSION

In conclusion, we believe that a key endeavour for future work should be to trace the evolution of SD across Arachnida more broadly, extending work that has thus far

predominantly been restricted to Araneae. For example, the frequency with which pedipalp SSD occurs across arachnids (seven out of 11 orders) may point towards an early origin within the group. Alternatively, given that arachnid pedipalps appear to be involved in numerous different courting, mating and other related tasks, and show many different types of SD, it is equally possible pedipalp dimorphism may have evolved independently several times. Such analyses will prove extremely informative with regards to the origin of SD in the group, but necessarily must overcome issues regarding phylogenetic uncertainty. In arachnids as a whole, there is little congruence between recent morphological and molecular phylogenies (*Sharma et al., 2014*; *Garwood et al., 2017*; *Giribet, 2018*); this issue is often replicated within individual arachnid orders. Furthermore, there is a general paucity of information on the phylogenetic relationships within smaller arachnid orders. For example, just one molecular phylogenetic study of Palpigradi has been published (*Giribet et al., 2014*). In Amblypygi, limited morphological phylogenies have been published (*Weygoldt, 1996*, *Garwood et al., 2017*) and no molecular phylogenetic study of the order as a whole has ever been conducted. Therefore, ideally future analyses of SSD should be accompanied by improved phylogenies, or else account for current uncertainty in phylogeny.

Furthermore, we note that basic data pertaining to the biology and life history of many arachnid orders are still lacking, particularly in the smaller groups. For example, information on courtship displays in Schizomida are limited to anecdotal evidence, and there is no published data on mating in Palpigradi. An improved understanding of ontogenetic scaling in the size and shape of arachnids is also a priority. In particular, the ability to better identify discrete ontogenetic stages and the onset of sexual maturity will prove useful, as dimorphism frequently becomes more pronounced beyond this point.

Future research efforts should also exploit recent advances in the fields of morphometrics, statistics, experimental physiology and biomechanics. Some progress has been made in this direction concerning Araneae SD: for example, recent studies have employed geometric morphometric to quantify shape dimorphism amongst *D. merlini* (*Fernández-Montraveta & Marugán-Lobón, 2017*). In contrast, potential shape dimorphism amongst the smaller arachnid orders is typically quantified using ratios of linear metrics (*Weygoldt, 2000*; *Vasconcelos, Giupponi & Ferreira, 2014*; *Santos, Ferreira & Buzatto, 2013*), and may therefore fail to capture finer-scale shape change between sexes. Furthermore, statistical hypothesis testing remains limited amongst the smaller orders. Whilst limited sample sizes are both frequent and undoubtedly a problem, other studies comprising a larger number of samples continue to eschew statistical testing, and further work is needed to statistically corroborate previously published qualitative observations. Finally, field and lab-based experimental studies are uncommon outside of spiders (*Moya-Laraño, Halaj & Wise, 2002*; *Grossi & Canals, 2015*). This work is, however, imperative, as an improved understanding of form-function relationships will provide further insights into the life history of both sexes, and the potential evolutionary drivers behind SD within arachnids.

## ACKNOWLEDGEMENTS

We thank the editor, Glaucho Machado and one other anonymous reviewer, whose comments greatly improved this manuscript.

### Funding

The authors received no funding for this work.

### Competing Interests

The authors declare that they have no competing interests.

### Author Contributions

- Callum J. McLean analysed the data, prepared figures and/or tables, authored or reviewed drafts of the paper, approved the final draft.
- Russell J. Garwood authored or reviewed drafts of the paper, approved the final draft.
- Charlotte A. Brassey authored or reviewed drafts of the paper, approved the final draft.

### Data Availability

The raw data are provided in the Supplemental Files.

### Supplemental Information

Supplemental information for this article can be found online at http://dx.doi.org/10.7717/peerj.5751#supplemental-information.

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
