# Peer review of "Sexual dimorphism in the Arachnid orders"

_PeerJ, doi:10.7717/peerj.5751_

## Round 0.1 · original submission · Major Revisions

I apologize for the delay in my response, but I waited long for a third review which unfortunately did not come, and wanted to go through the manuscript one more time to able to give the best possible recommendations on how to proceed. You provide a broad and interesting review on dimorphism in arachnids, which I feel is long overdue. For this reason I would really like to see it published. However, there are some crucial points I would like you to address before publication.

The main points are:

Goal of the study: it is not entirely clear if the aim of the manuscript is to be a systematic review or an overview of sexual dimorphism in arachnids with some selected case studies (see comments by reviewers 1 and 2). Both approaches have their merits, but I agree with reviewer 1 that the systematic review would be a more relevant contribution.

Scientific reproducibility: Currently, it is not entirely clear how the listed studies/reported patterns were chosen (see also reviews 1 and 2). You briefly describe your approach to find studies, but it is hard to verify as you do not provide a table with included studies or how many taxa you found publications on, etc. Furthermore, on several occasions statements are made like “most-researched group”, “most documented instances are qualitative” and “SD is prevalent” without providing references. Reviewer 1 pointed out that several cases of sexual dimorphism in arachnids were not discussed. Furthermore, mention that research focused on Order Araneae, but you do not back this up with a lot of references nor do you summarizes the main results from those studies (see comments by reviewer 2). I agree with the reviewer 1 that when performing a systematic review the PRISMA (Preferred Reporting Items for Systematic reviews and Meta-Analyses) protocol (http://www.prisma-statement.org/) would be beneficial. Such a protocol would allow the readers to understand how the literature search has been conducted, how many papers have been initially found, how many have been discarded, and how many papers were ultimately included in the review allowing both qualitative and quantitative information. I agree with Reviewer 1 that it would make your paper much stronger if the following topics could be addressed: (1) how many species were detailed studied in each order?; (2) For how many species do we have statistical data on sexual dimorphism?; (3) What is the percentage of species in each order for which we have reports of sexual dimorphism on body size, legs, chelicerae, pedipalps, etc.? All this information could be presented in a table which lists studies, species/genera that have been tackled, which patterns were observed, how they were documented (qualitative versus quantitative, if quantitative – which method and significant or not, etc.) and to which they were attributed (behavior, sexual selection, etc.,)

Structure: For each group; you discuss the phylogeny, types of dimorphism and the potential relationship with mode of life, behavior, sexual section and other peculiarities. It would help if would introduce subtitle for these sections for each group in the same way.

Impact of mode of life, behavior, natural or sexual selection: you sometimes briefly discuss the potential relationship between types of dimorphism and various aspects like mode of life, behavior and the potential impact of selection. It would benefit it these would be done for all groups in the same extent (see above) with particularly reference to how males and females use sexually-dimorphic structures and it might also be interesting to even expand how selective pressures may have promoted the evolution of some types of sexual dimorphism (see comments by reviewer 1). This could be done by dedicating more space in the discussion to explore the evolution of sexual dimorphism, which may be the result of both natural and sexual selection acting on males and/or females as suggested by reviewer 1. In this context, I feel it would also be good to already discuss and homogenize some aspects in the sections on particular groups, so in the end the focus could lie more on the differences and relative contributions of these factors among arachnids.

Phylogeny: I understand the phylogeny of arachnids is still in a state of flux. However, there are some groups which are well supported, while other jump around depending on the methods/data used (Giribet, 2018). I feel it would benefit the reader if you would actually have a figure with potential relationships between taxa in your paper – well supported ones could be resolved, while not well-supported ones could be polytomies. For this purpose, you could for example use Giribet (2018), although another one would also be fine. By using the Giribet (2018), I was striking how groups which are thought to be closely related, have quite similar patterns based on your table. Probably not the main point of the review, but it could at least also be briefly mentioning if dimorphism has also been observed or discussed in extinct arachnids belonging to extant or extinct groups.

Drawings: I think the drawings of SSD patterns are a great idea – I am just not sure if they need to be that big. I think have all next to one another and/or potentially combined with the phylogeny suggested above would be even more informative. If you want to keep the large versions to – they could potentially be used to show how many studies have documented this patterns for particular appendages and/or potentially graphically in which direction they might differ – I know female-biased or male-based convey the same information, but somehow it might be more informative if arrows are used with various thickness depending on the prevalence? These are just suggestions.

Terminology: Please use clear terminology throughout your manuscript (see comments by reviewer 2). You seemingly differentiate between SD and SSD, but sometimes these terms are confused (e.g., SSD in Acari´s glandular system on line 208; see comments by reviewer 2). See also further comments by reviewer 2.

In addition to these and other points raised by the reviewers, please also address the following:

Line 50: “particularly problematic”: sounds a bit harsh – I am sure those authors focused on this dataset for a reason
Line 58-59: it is interesting that both can be at work in different parts of the body. Such patterns can potentially only be revealed by using geometric morphometrics (e.g., https://peerj.com/articles/3617/)
Line 66: I guess you mean “Opiliones”
Line 71: I agree that it is timely and important
Line 139: survey methodology: please use “PRISMA” framework and report all studies and how many species or higher taxa were studied, etc.
Line 185: do you mean present or reported – you need to back this up with reference(s) and ideally a list with how many study you found reporting it
Line 204: the lack of information is maybe a bit strong – maybe relative lack of information would be more appropriate?
Line 424: “The” instead of “he”
Line 430: “might be” might be more appropriate then “is”
Line 467:”cheliceral” instead of “chelceral”
Line 478: “that it” instead of “the it”
Line 525: you need reference to back of such a statement “most-researched arachnid group”
Line 535: “No consensus” instead of “N consensus”
Line 564: SD in Ricinulei are qualitative => it would be appropriate to have a table for all groups tabulating studies, which taxa and how they have been studied.
Line 643-644: what does it mean to show “little to no SSD”; please define what counts as “little”
Line 754: “in total size is yet to be addressed” – is there no way to do some of these corrections a posteriori by correcting for allometric relationship between those
Line 687: “above difficulties”; do you mean the difficulties discussed in the beginning – it might be appropriate to refer to the exact lines or at least the exact section in that case.
Line 747-748: “numerous studies report SD in the chelicerae” => you need to provide reference or refer to a table where these are listed.
Line 882: considering the poor study in some groups, it might be more appropriate to state that it is documented in mites, etc. rather than present => absence of proof is not necessarily proof of absence
Line 919: Giribet 2018 would also be an appropriate reference in this context

·

Basic reporting

See "General comments for the authors".

Experimental design

See "General comments for the authors".

Validity of the findings

See "General comments for the authors".

Additional comments

In this manuscript, the authors revise the forms of sexual dimorphism in arachnids, a major group of arthropods with an enormous diversity of body plans and behaviors. Although there is scattered information on sexual dimorphism of isolated species and also of some specific clades, there is no comprehensive review of the subject spanning the whole class. As someone that works with arachnids and is interested in sexual selection, I thus think the manuscript is very welcome. However, I have three major concerns about the manuscript. These three concerns are more or less interconnected and I will elaborate each of them below.

1) It is not clear to me if the manuscript intends to be a systematic review or simply an overview of sexual dimorphism in arachnids in which the authors select some emblematic examples. Both goals are valid, but it would be important to properly explain to readers what kind of review authors want to provide. I think this point is important because there many cases of sexual dimorphism in arachnids that were not addressed in the manuscript. In what follows, I mention examples for some orders.

1.1) Acari

- I am conducting a huge review of parental care in arthropods and, based on my own experience, I know that is very hard to compile information on Acari. Although there are several books covering different aspects of their biology, detailed reviews on specific subjects are basically non-existent. In the manuscript, the authors provide general data on sexual dimorphism in Acari, focusing on oribatids. In fact, this is the most tractable group within the Acari because it is more intensively studied than other groups. However, even in oribatids there are some reports of sexual dimorphism that were not mentioned in the manuscript. For instance, there is at least one case of sexual dimorphism in leg length in oribatids. In Ameronothrus lineatus, adults are sexed by the gender-specific size difference and also the longer legs relative to body size of the males. Please see: "The biology and life history of arctic populations of the littoral mite Ameronothrus lineatus (Acari, Oribatida)" -- G. Søvik (Experimental and Applied Acarology 34:3-20, 2004).

1.2) Araneae

- There is a very conspicuous case of sexual dimorphism in the linyphiid Oedothorax gibbosus. Males have a huge gland on the cephalothorax that provides a nuptial secretion to females. Curiously, the presence of this gland is also intrasexually dimorphic, so that a male morph without the gland is also found in natural populations. Please see: "Male dimorphism in Oedothorax gibbosus (Araneae, Linyphiidae): a morphometric analysis". -- S. Heinemann & G. Uhl (Journal of Arachnology 28:23–28, 2000).

- Among the Mygalomorphae, there are several cases of sexual dimorphism in the length of urticating setae. Please see review in: "Morphology, evolution and usage of urticating setae by tarantulas (Araneae: Theraphosidae)" -- R. Bertani & J.P.L. Guadanucci (Fortschr. Zool. 30:403–418, 2013).

- Elongated legs in males may have different functions and I think these different functions could be better explained and explored in the manuscript. In some species, males use elongated legs in intrasexual contests. Leg length is probably used in the evaluation of the opponent as an indicator of his size and strength. Thus, the main selective pressure leading to leg elongation in these species is intrasexual selection. In other species, legs are used as sexual display to attract or court females and the main selective pressure is intersexual selection. Although some examples of leg elongation probably promoted by intersexual selection were mentioned in the manuscript, I think the authors should provide more information and examples of leg elongation probably promoted by intrasexual selection (i.e., male-male contests). Example: "Behavior and ecology of four species of Modissimus and Blechroscelis (Pholcidae)" -- W. G. Eberhard & R. D. Briceño (Rev. Arachnol. 6:29-36, 1985).

1.3) Opiliones

- There is a well-reported case of sexual and intrasexual dimorphism in the length of the second pair of legs in the harvestman Serracutisoma proximum. The second legs are used as whips in male-male fights for the possession of reproductive territories. This example has not been mentioned in the manuscript, and considering that we know much more about the biology of S. proximum than we know about Phalangium opilio, I think the attention devoted to this later species is unbalanced in the manuscript (which also includes a figure about sexual dimorphism in P. opilio). Please see: "Resource defense polygyny shifts to female defense polygyny over the course of the reproductive season of a Neotropical harvestman" -- B. A. Buzatto & G. Machado (Behavioral Ecology and Sociobiology, 63:85-94, 2008). / "Conditional male dimorphism and alternative reproductive tactics in a Neotropical arachnid (Opiliones)". B. A. Buzatto et al. (Evolutionary Ecology, 25:331-349, 2011) / "Male dimorphism of a neotropical arachnid: harem size, sneaker opportunities, and gonadal investment" -- R. Munguía-Steyer et al. (Behavioral Ecology, 23:827-835, 2012) / "A sexual network approach to sperm competition in a species with alternative mating tactics" -- D. G.Muniz et al.(Behavioral Ecology, 26:121–129, 2015) / "Experimental limitation of oviposition sites affects the mating system of an arachnid with resource defence polygyny" -- D. G. Muniz & G. Machado (Animal Behaviour, 109:23-31, 2015) / "Devoted fathers or selfish lovers? Conflict between mating effort and parental care in a harem-defending arachnid" -- L. M. Alissa et al. (Journal of Evolutionary Biology, 30: 191-201, 2017).

- There is a review of intrasexual dimorphism in harvestmen: "Male dimorphism and alternative reproductive tactics in harvestmen (Arachnida: Opiliones)" -- B. A. Buzatto & G. Machado (Behavioural Processes, 109:2-13, 2014). Although the focus of this paper is on intrasexual dimorphism, I think it can provide useful information for your manuscript on sexual dimorphism. In general, minor males are similar (but not identical) to females, whereas major males have exaggerate traits that make them different from both minors and females. The paper provides a table with a review of the known cases of intrasexual dimorphism in harvestmen and also indicates which cases are supported by statistical evidence and which cases are only "observational".

- In the family Neopilionidae, there are several cases of extreme male morphologies related to cheliceral size and shape. In a recent paper on this family, there is the first description of male trimorphism in arachnids. As I said above, the same traits that are intrasexually dimorphic are sexually dimorphic in neopilionids. Please see: "Multiple exaggerated weapon morphs: a novel form of male polymorphism in harvestmen" -- C. J. Painting et al. (Scientific Reports, 5:16368, 2015).

- In a recent paper, there is a macroecological analysis of the magnitude of sexual dimorphism in harvestmen. We first showed that the length of the breeding season in harvestmen as a whole is strongly determined by the number of months with proper climate conditions (mean temperature above 5 °C and high precipitation). The length of the breeding season, in turn, strongly influences the type of mating system of each species: when the breeding season is very short, males rely mostly on scramble competition polygyny; when the breeding season is long, males rely mostly on resource defense polygyny. Finally, the type of mating system is correlated with the magnitude of sexual dimorphism in harvestmen: in species with scramble competition males do not bear weaponry and are usually smaller than females; in species with resource defense, males bear weaponry and are usually larger than females. I really think this paper provides nice information for your review. Please see: "Macroecology of sexual selection: a predictive conceptual framework for large-scale variation in reproductive traits" -- G. Machado et al. (The American Naturalist, 188:S8-S27, 2016).

2) My second criticism is tightly related to the first one: based on the information provided by the authors, it seems to be impossible to replicate the result of the literature search. In the last years, there is growing concern about reproducibility of scientific research and people working with any kind of systematic review proposed the PRISMA (Preferred Reporting Items for Systematic reviews and Meta-Analyses) protocol (http://www.prisma-statement.org/). This protocol allows the readers to understand how the literature search has been conducted, how many papers have been initially found, how many have been discarded, and how many papers were ultimately included in the review. Following this protocol has a great advantage: the authors can provide both qualitative and quantitative information for the readers. I am not proposing the author conduct a meta-analysis, but they can add numbers to provide a thorough general picture. The following topics could be addressed: (1) how many species were detailed studied in each order?; (2) For how many species do we have statistical data on sexual dimorphism?; (3) What is the percentage of species in each order for which we have reports of sexual dimorphism on body size, legs, chelicerae, pedipalps, etc.? In the current format, we have an interesting collection of examples and a poorly-supported attempt to detect broad patterns. I am sure that numbers would make the review much more useful and appealing to a broad audience.

3) Overall, there is extensive information on the types of sexual dimorphism in different arachnid orders, but the functional meaning of these types of dimorphisms is not properly explored in the manuscript. According to my view, the review could be greatly benefitted from more behavioral information related to how males and females use sexually-dimorphic structures. Even for poorly-studied groups there are behavioral data for some species describing how males use their sexually-dimorphic structures. This is the case of pseudoscorpions, which are nicely explored by the authors in the manuscript. For other groups, such as whip-spiders and harvestmen, there is also information on the behavioral role of sexually-dimorphic male traits that could be better explored in the review.

Moreover, the selective pressures that may have promoted the evolution of some types of sexual dimorphism could be expanded. Although there some brief comments on this subject along the manuscript, they are not evenly distributed among the orders (and here I am only talking about the more intensively studied groups, such as spiders, scorpions, harvestmen, pseudoscorpions, and perhaps camel-spiders and whip-spiders). As the authors state in the introduction, the evolution of sexual dimorphism may be the result of both natural and sexual selection acting on males and/or females. In my opinion, the manuscript would be more useful and informative if the authors devote more space in the discussion to explore the role of these selective pressures. More specifically, the following topics could be addressed: (1) Is there any evidence that niche partitioning promotes sexual dimorphism in arachnids? If so, which sex seems to be modified and why?; (2) Is there any evidence about fecundity selection acting on females in the different orders? I am sure that there are correlations between female size and fecundity for many different species; (3) When males have sexually dimorphic structures, are they used mainly as weapons or ornaments? In other words, what selective pressure seems to be more pervasive, intra- or intersexual selection?; (4) Scramble competition seems to be the most common mating system in arachnids and, according to the theory, sexual selection should favor male traits that increase the chances of finding a receptive female. These traits would include longer legs and more developed sensory structures (Andersson, 1994: book "Sexual selection"). Does this prediction hold for arachnids as a whole?

Besides the three major concerns mentioned above, I also have numerous small comments and suggestions, which are presented below.

I hope my comments help the authors to improve the manuscript. In case of any doubt, please contact me.

Glauco Machado
Department of Ecology
University of Sao Paulo
E-mail: glaucom@ib.usp.br

= = = = = = = = = = = = = = = = = = = = = =

Minor comments/suggestions:

Line 40: "(Shine, 1989; Andersson, 1994)"

Lines 40-46: please consider the following suggestion:

"Documenting and describing the occurrence of sexually dimorphic traits can provide important insights into the evolution of morphology. Amongst vertebrate groups, for instance, the occurrence of SD is well-documented. In mammals, SD has been compared in 1370 species, representing around 30% of known mammalian species (Lindenfors, Gittleman & Jones, 2007). Datasets of similar size have been used to quantify SD in reptiles (Cox et al. 2007) and birds (Owens & Hartley, 1998)."

Line 65: "...asymmetry (Proctor, 2003), extreme size..."

Line 84: "...across eleven modern..."

Lines 103-112: please see the following paper: Kilmer JT, Rodríguez RL, 2015. Do structures with sexual contact functions evolve negative static allometries? A case study with the harvestman Leiobunum vittatum (Opiliones Sclerosomatidae). Ethol. Ecol. Evol. 29:64–73.

Line 121: "(e.g. Moya-Laraño et al. 2002; Foellmer & Moya-Larano 2007; Grossi & Canals 2015)"

Line 133: "...on the posterior part of the opisthosoma in schizomids..."

Line 166: "Garwood et al.’s (2017)"

Line 170: "...with this clade being the sister group..."

Line 174: "(e.g. Behan-Pelletier & Eamer 2010; Behan-Pelletier 2015a, b)"

Lines 175-176: please merge these two paragraphs.

Line 185: "...some mite species."

Line 198: "Setal arrangement also varies between sexes..."

Lines 200-203: "Male pedipalps are enlarged relative to female conspecifics and have branches unseen in females in some species of Astigmata, and in the most extreme cases appear antler-like (Proctor, 2003). Chelicerae are also enlarged in some male feather mite species (Proctor, 2003)." -- Thus, in this group, pedipalps and chelicerae are not fused to each other forming the gnathosoma, right? Please explain it.

Line 221: "Research into SD among mites and ticks has thus far been..."

Line 281: in the arachnological literature, the expression "true spiders" is used mostly to the Araneomorphae and not to the order as a whole.

Line 282: "(Platnick & Raven, 2013)" -- I suggest to change this citation for the most recent version of the World Spider Catalogue (https://wsc.nmbe.ch/).

Line 284: "...and the ability to produce silk" -- at least two other arachnid groups contain species that produce silk: pseudoscorpions and mites. Thus, you should to re-phrase this sentence.

Line 288: "Spiders are typically characterised..."

Line 292: are you really referring to the "Tetragnathidae" or to the "Nephilidae"?

Lines 293-294: another well-reported case of reversed SSD in spiders occurs in the genus Allocosa from Uruguay. In species of this genus, males are larger than females and there are several behavioral traits that indicate sex-role reversal.

Lines 306-318: I suggest to remove this paragraph because pedipalps in male spiders are sexual organs, thus you are talking about primary and not secondary sexual dimorphism. If you decide to mention all cases of primary sexual dimorphism in all arachnid orders, you will need to talk about spermatophore and penis morphology. As far as I understand, this is not the main goal of the manuscript. Thus, for the sake of consistency, you should not include sexual dimorphism related to pedipalp morphology in spiders. The only exception, according to my view, is the example of Allocosa males, which have sexually dimorphic digging structures on their pedipalps.

Line 350: "conflict" -- I think the best word here is "contest". Please change it throughout the entire manuscript.

Line 351: " conflict" -- I suggest to change this word for "male-male competition" or "intrasexual selection".

Line 360: "and" -- non-italic.

Line 387: "Parisitiformes"

Lines 424-425: "The metric..."

Lines 427: "Cranaidae and Oncopodidae"

Lines 456-457: "...suggesting that chemical secretions may also play a role in warding off rival males" -- Although it is possible, I have never seen males releasing secretion during intrasexual fights.

Line 465: " (Metasarcidae, Cranaidae and Oncopodidae..."

Line 467: "During contest, males..."

Line 476: the word "ornaments" is inadequate because there is no empirical evidence supporting that sexually-dimorphic structures in harvestmen function as ornaments. Most of them unequivocally function as weapons.

Lines 481-482: "Identifying a reliable proxy for overall body size and statistically testing SSD should be a priority" -- According to my personal experience, there is no general proxy of overall body size in harvestmen. Perhaps the best candidate would be the width of the cephalothorax, but this body segment is highly fused with other body segments. Thus, in many groups, it is difficult to recognize the cephalothorax and measure it.

Lines 507-598: "It is not uncommon, for example, to find both strong male-biased and female-biased SSD in claw size within a genus (see figure 9, Zeh, 1987b)" -- Please note that the phylogenetic studies of pseudoscorpions begun only in the late 1990's. I would not be surprised if the species used in the study by Zeh (1987b) now belong to different genera. Please try to check this information in the excellent website by Mark Harvey (http://www.museum.wa.gov.au/catalogues/pseudoscorpions).

Line 515: " (Weygoldt, 1966, 1969)"

Lines 531-532: in the section on Palpidradi you say: "They are also the least speciose order, with 78 extant species (Harvey, 2003)" (line 379-380). Here you say that "Ricinulei, or hooded tick spiders, are a small arachnid order comprising only 58 described species (Prendini, 2011)". Based on the numbers you presented, Ricinulei is the least speciose order.

Line 535: "No consensus exists..."

Line 555 and 678: "leg play" -- What does it mean? Is it a type of copulatory courtship?

Line 558: "...and longer (Pittard, 1970) in male..."

Line 603: "...Furthermore, the male pedipalp..."

Line 657: "In Heterometrus laoticus the telson is found..."

Lines 658-659: "(Booncham et al., 2007)"

Line 660: "Other structural modifications can be found (?) in males of Anuroctonus

Line 660: "Phaiodactylus"

Line 685: "sp.)" -- non-italics.

Line 763: "manus" -- What is it? Please explain.

Line 821: "near the abdomen" -- Please be more specific.

Line 839: "...and the patella..."

Line 842: "tibial apophysis" -- On the pedipalp? Please make it clear.

Line 845: "(Gravely, 1915; Yoshikura, 1973; Weygoldt, 1972, 1978; Rajashekhar & Bali, 1982)"

Line 848: "T. indicus" ("and" -- non-italics)

Line 858: " Klingel, 1963; Weygoldt, 1971, 1972"

Line 889-890: "...female scorpions may engage in sexual cannibalism of males (see Peretti et al. 1999); this could hypothetically to lead to the rare male-biased SSD within this group" -- According to Peretti et al. (1999), "no post-mating cannibalism was ever observed in [scorpions]". Thus, any hypothesis to explain sexual size dimorphism based on the assumption of sexual cannibalism should be revised or eliminated from the manuscript.

Line 892: "hypothesised in some species of the spider family Linyphiidae (Lang, 2001)" -- This example was not mentioned in the topic about spiders.

Line 900: "...leg length, shape, armature..."

Line 917: "...uncertainty. In arachnids..."

FIGURES:

- I suggest to merge figures 1, 2, 4, 4, 6, 8, 10, 11, 13, 15 and 16 into a single plate for comparative purposes. Please see the suggestion below:

"Fig. 1 – Patterns of sexual dimorphism in arachnids: (a) Acari, (b) Amblypygi, (c) Araneae, (d) Palpigradi, (e) Opiliones, (f) Pseudoscorpiones, (g) Ricinulei, (h) Schizomida, (i) Scorpiones, (j) Solifugae, and (k) Uropygi. In all schemes body parts coloured red indicate male-biased SSD, green indicates a female bias, and purple mixed sex bias. Legs are numbered 1-4, chelicerae are marked “C” and pedipalps are marked “P”; male (♂) or female (♀) symbols denote sexual size dimorphism in overall body size."

- If you accept the suggestion above, please remove the text from lines 146 to 151 (this information is now presented in the legend of Fig. 1) and from lines 1469 to 1472 (this information is already presented in the main text).

- Fig. 17 is in fact a table.

Reviewer 2 ·

Basic reporting

The manuscript entitled “SEXUAL DIMORPHISM IN THE ARACHNID ORDERS” is aimed to provide an overview of the research to date on sexual dimorphism in individual arachnid orders, using these data to analyse trends driving the evolution of sexual dimorphism across Arachnida.

This is a generally well-written paper, with an appropriate use of academic English throughout, except for small typos (see comments to the authors).

The paper does not always include an adequate framework where to place its aims, and some relevant references are lacking. For example, the authors emphasize that spiders are the only arachnid group in which some broad revision of SSD has been done. However, the section devoted to the Order Araneae is extremely limited. Authors summarise main findings in this group by saying that research has probed the degree to which spiders follow Rensch’s Rule (L. 57), but they do not provide a single reference to support this view, nor do they document that the opposite, i.e., that spiders do not fulfil predictions from the rule, is generally accepted to be true.

Structure generally conforms PeerJ standards. However, having to review patterns of sexual size and shape dimorphism in such a morphologically diverse taxonomical group requires a huge amount of information and a large extension, making the paper excessively long and fragmented.

Figures are relevant to illustrate the morphological diversity of the analysed taxa.

No new raw data are provided, as the paper only reviews already published data.

Experimental design

This is a literature review article within the scope of PeerJ.

The main contributions of the paper are supposedly related to the range of taxonomic groups included, the use of type II regression to analyse patterns of sexual size dimorphism and the choice of appropriate estimates of body size. However, the real contributions of this paper to filling these gaps is questionable. Actually, the paper mostly summarises common trends of variation described up to date and lacks any new proposal for data analysis or size estimates. For example, figure 3 represents results from a type II regression analysis, but there is not a single reference in the text confronting this with alternative techniques.

The use of methodological terms is obscure. One of the authors’ claims is that shape should be considered together with size in the analysis of sexual dimorphism. However, throughout the paper, authors mostly refer to sexual size dimorphism, and they use size and shape as two interchangeable terms. For example, they talk about sexual size dimorphism in Acari’s glandular system (L. 208), when referring to the arrangement of the dermal porose areas.

The Methods section is extremely short and undetailed. Particularly, the authors give no clue on the criteria used to select the papers eventually included or excluded from their revision.

Validity of the findings

Spiders have increasingly become a model group in the evolutionary analysis of sexual size dimorphism, whereas information about other arachnid groups is scarce. Enlarging the range of taxa analysed looks extremely promising. However, at its present state, this paper mostly describes patterns of variation and does not make any new or original contribution to the analysis of their evolutionary origins. Probably, the research question is not appropriately addressed, and the paper should be better addressed to a taxonomically oriented audience.

Though this is not a primary research, a lot of information needed to support the author’s claims is lacking. Particularly, there is a lack of essential information about ecology and phylogeny, limiting the paper scope in the analysis of evolutionary trends. Indeed, one of its main conclusions is that hypotheses about the evolutionary history of arachnid sexual dimorphism are limited by the uncertainty in arachnid phylogenetic relationships.

Conclusions are not always clearly drawn from the data. For example, the main conclusion of the paper is the driving role of fecundity selection (L 885) when explaining the general F>M trend of sexual size dimorphism across arachnids. However, only two arachnid orders are acknowledged to follow this pattern when referring to body size, whereas the opposite (relatively larger males, M>F) is true across the Class, as shown in Table 1.

Additional comments

L. 84: typo (elven)

L. 91-102: there seems to be a confusion of terms. Assessment of Sexual dimorphism in vertebrates is diverse, and it is uncertain to say that it is generally based on body mass. Similarly, there is an agreement to use carapace width as an indicator of spider body size, whereas the width/length ratio of the carapace should be used as a shape, and not size, estimator

L. 97 typo (thought to indicative)

---

## Round 0.2 · Minor Revisions

Thank you for addressing most of our suggestions. The partial restructuring and the addition of a phylogeny have made the manuscript easier to follow and of even high value as a review of dimorphism across arachnids. Both me and the reviewer have some additional small comments/suggestions which i would like you to take care of before publication. These are listed in the annotated pdfs. I am looking forward to receiving and the publication of your revised manuscript.

·

Basic reporting

The revised version of the manuscript has been substantially improved and all my major comments were properly addressed by the authors. I have several small comments and all of them were directly inserted in the text. Please consult the attached file.

Experimental design

Please consult the attached file.

Validity of the findings

Please consult the attached file.

Additional comments

I'm really glad to see how much energy you put in the revised version of the manuscript. I think the text has been greatly improved and I'm sure that this review will be an important contribution to the field and a highly cited paper. Congratulations!

---

## Round 0.3 · accepted · Accept

Thank you addressing all but one of final suggestions. I also appreciate the explanation for keeping figures charting trends of SSD in each order separately and adding a complete of all the SSD trend figures in the supplementary material. Looking forward to the publication of your manuscript.

#